**Review Article**

# Overflow metabolism in bacterial, yeast, and mammalian cells: different names, same game

Thomas Gosselin-Monplaisir [ID][1], Brice Enjalbert [ID][1], Sandrine Uttenweiler-Joseph [ID][1], Jean-Charles Portais [ID][1,2,3], Stéphanie Heux [ID][1] & Pierre Millard [ID][1,2 ✉]

## Abstract

**Overflow metabolism refers to the widespread phenomenon of cells excreting metabolic by-products into their environment. Although overflow is observed in virtually all living organisms, it has been studied independently and given different names in different species. This review highlights emerging evidence that overflow metabolism is governed by common principles in prokaryotic and eukaryotic organisms. We examine the similarities and specificities in the structure, function, and regulation of overflow pathways in bacterial, yeast, and mammalian cells, with a focus on model species and common by-products. Our reinterpretation of previous findings points to the existence of universal principles governing overflow fluxes. We also emphasize the need to reconsider the roles of overflow metabolites, not as cellular stress-inducing toxic waste, but as nutrients and regulators, influencing metabolism at both cellular and community levels, often to the benefit of the producing cells. Finally, we review prevailing theories of overflow metabolism and explore avenues toward a potential unified theory of overflow. This review offers fundamental insights into this widespread metabolic process and proposes a conceptual foundation for future research.**

**Keywords** Acetate Overflow; Crabtree Effect; Warburg Effect; Central Metabolism
**Subject Category** Metabolism

## Introduction

Overflow metabolism refers to the phenomenon whereby cells excrete metabolic waste products in conditions where theoretically, they could completely oxidize nutrients such as glucose. This phenomenon is observed in virtually all living organisms. First reported by Pasteur in 1857 (although later named the Crabtree effect) for ethanol production by the yeast *Saccharomyces cerevisiae* (Pasteur, 1860), Harden observed the production of acetate by the bacterium *Escherichia coli* in 1901 (Harden, 1901), while the production of lactate by mammalian cells was elucidated in the

1920s and named the Warburg effect in cancer cells (Warburg and Minami, 1923). This ubiquitous overflow phenomenon represents a loss of carbon and energy for cells because the produced compounds are first excreted instead of being directly used for growth. Moreover, since overflow by-products inhibit cellular growth, they are traditionally considered toxic waste (Luli and Strohl, 1990; Brown et al, 1981; Lao and Toth, 1997). Understanding why cells produce self-inhibiting molecules instead of using glycolytic substrates more efficiently remains an intriguing question that has driven metabolic studies for more than a century.

Overflow metabolism also underlies a wide range of applications (Kiefer et al, 2021; Zhang et al, 2022; Rabinowitz and Enerbäck, 2020; Van Der Hee and Wells, 2021). Healthcare-relevant overflow by-products include lactate produced by host cells and short-chain fatty acids (SCFAs, e.g., acetate, propionate) produced mainly by the gut microbiota (Van Der Hee and Wells, 2021). Lactate, a circulating metabolite with a long-established reputation as a detrimental waste product, has in recent years been recognized as a major carbon and energy shuttle between organs (Rabinowitz and Enerbäck, 2020), and dysregulation of lactate metabolism is observed in cancer, obesity, and other diseases (Brooks, 2018). Similarly, dysregulation of microbiota-derived SCFAs metabolism leads to microbiome-related disorders (Van Der Hee and Wells, 2021). Understanding overflow metabolite dynamics, as well as the roles and responses of host cells and of the gut microbiome to these dynamics, could significantly improve healthcare. In biotechnology, overflow metabolites are potential resources in renewable feedstocks (Kiefer et al, 2021; Zhang et al, 2022); however, their presence or accumulation in bioprocesses diminishes productivity by inhibiting growth and diverting carbon fluxes that could otherwise be used to synthesize biomass or valuable compounds (De Mey et al, 2007; Torres et al, 2018; Zhang et al, 2022). Alternatively, the co-utilization of overflow metabolites with glycolytic substrates may mitigate growth inhibition while recycling by-products (Raamsdonk et al, 2001; Mulukutla et al, 2012; Millard et al, 2023; Nam et al, 2024). Despite intensive efforts to limit their accumulation or increase their utilization, overflow by-products still pose a significant challenge in biotechnology (Zhang et al, 2022; Torres et al, 2018).

Despite its universality, overflow metabolism has largely been studied independently in different organisms. This review outlines the similarities and specificities of overflow pathways in bacterial, yeast, and mammalian cells in terms of their topology, function,

[1]TBI, Université de Toulouse, CNRS, INRAE, INSA, Toulouse, France. [2]MetaToul-MetaboHUB, National Infrastructure of Metabolomics and Fluxomics, Toulouse, France. [3]RESTORE, Université de Toulouse, Inserm U1031, CNRS 5070, UPS, EFS, Toulouse, France. ✉E-mail: pierre.millard@insa-toulouse.fr

and regulation. We particularly focus on *E. coli*, *S. cerevisiae*, and CHO cells, the three major unicellular chassis in biotechnology. We reevaluate previous findings highlighting strong functional similarities between these three model species, which point to the existence of universal principles governing overflow fluxes. We also consider how the presence of overflow products in the environment reshapes the physiology of the producing cells, with significant implications for their carbon, redox, and energy metabolism. While overflow products have long been considered toxic, or stress factors, emerging evidence suggests they should instead be regarded as valuable nutrients and beneficial global regulators. Finally, we examine prevailing theories on overflow metabolism and discuss how to move forward toward a unified theory of overflow. This review provides fresh insights into this ubiquitous yet elusive metabolic process and paves the way for more efficient strategies to exploit overflow metabolism in applications ranging from biotechnology to healthcare.

## The classical view of overflow metabolism emerging from diauxic growth

The preferred carbon sources of most living organisms are glycolytic substrates (Görke and Stülke, 2008), which are taken up and converted into pyruvate through glycolytic pathways (Romano and Conway, 1996), including the canonical Embden–Meyerhof–Parnas pathway (i.e., glycolysis), the Entner–Doudoroff pathway, and the pentose phosphate pathway. Pyruvate can undergo further oxidation into $CO_2$ through the tricarboxylic acid (TCA) cycle and derivative pathways, sustaining the biosynthesis of cellular building blocks and providing energy for growth. Alternatively, pyruvate may undergo incomplete oxidation into different compounds, which are then excreted and accumulate in the medium (Fig. 1A). In yeasts such as *S. cerevisiae* for example, pyruvate is converted into ethanol through pyruvate decarboxylase (Pdc) and various alcohol dehydrogenases (Adh); in mammalian cells, pyruvate is converted into lactate via different lactate dehydrogenases (Ldh); and in bacteria such as *E. coli*, pyruvate is converted into acetate through pyruvate dehydrogenase (Pdh), phosphate acetyltransferase (Pta) and acetate kinase (AckA). While acetate production involves the production of ATP, ethanol and lactate production enable the reoxidation of NADH into $NAD^+$, which can sustain glycolysis. Acetate, lactate, and ethanol are then excreted into the environment by diffusion and various transporters.

Overflow products are then reconsumed after depletion of glycolytic substrates (Fig. 1B). This phenomenon, known respectively as the acetate switch, ethanol switch and lactate switch, is observed in bacteria (Wolfe, 2005), yeasts (Meyenburg, 1969), and mammalian cells (Ozturk et al, 1997). In *E. coli*, glucose exhaustion suppresses catabolite repression, leading to the expression of acetyl-CoA synthetase (Acs) (Wolfe, 2005), which converts acetate back into acetyl-CoA. Similarly, in *S. cerevisiae*, upon glycolytic exhaustion, the gene encoding for Adh2 is derepressed and ethanol is converted into acetaldehyde and ultimately acetyl-CoA (Mohd Azhar et al, 2017). In mammalian cells, lactate reconsumption is thought to be caused by LdhC expression (Hartley et al, 2018; Torres et al, 2018).

Diauxic growth has significantly shaped the classical perspective that specific enzymes are responsible for the production of overflow by-products during glycolytic growth, while other enzymes are responsible for their utilization only after depletion of glycolytic substrates (Meyenburg, 1969; Ozturk et al, 1997; Wolfe, 2005). Substantiating this perspective, when catabolite repression is partially impaired or weakened, all enzymes are expressed, and both the producing and consuming pathways operate concurrently (Peebo et al, 2014). In this situation, the rate of overflow product accumulation (i.e., the net contribution of both pathways) is notably reduced.

## Control of overflow fluxes is shared between overflow pathways, glycolysis, and the TCA cycle

While the topology of overflow pathways has long been established for many organisms (Romano and Conway, 1996), the factors determining overflow fluxes have only recently become clearer. A common pattern has emerged from metabolic control analysis, with the development of experimental or computational methods to quantify flux control (Fell, 2005). The typical approach involves measuring the flux response to a modulation of the activity of a given reaction (i.e., $V_{max}$). As outlined in this section, various experimental strategies have been used to modulate pathway activity, such as modulating the expression of a specific enzyme, iteratively deleting isoenzymes catalyzing a given reaction, using inhibitors, and employing different bioprocess strategies.

Early efforts to control the production of overflow products focused primarily on modulating the activity of enzymes catalyzing their biosynthesis (Fig. 2). For example, deleting Adhs in yeast was found to reduce ethanol production (Smidt et al, 2012), and in *E. coli*, progressively increasing *pta* expression using an IPTG-inducible promoter was found to increase acetate accumulation (Sun-Gu and Liao, 2008). Indirectly modulating the activity of these enzymes via transcriptional (Schuurmans et al, 2008), translational (De Mets et al, 2019), or post-translational regulators (Castaño-Cerezo et al, 2014) was observed to produce the same metabolic response. Alternatively, artificially increasing the activity of enzymes with increased expression during the by-product utilization phase, such as Acs in *E. coli* (Peebo et al, 2014) or LdhC in CHO cells (Hartley et al, 2018; Torres et al, 2018), has also been shown to reduce by-product accumulation. These results are consistent with the negative control of overflow fluxes by the enzymes involved in by-product utilization.

Other controlling pathways have also been identified. A notable reduction in by-product excretion has been observed in these organisms when the glycolytic flux is reduced by bioprocess or metabolic engineering (Fig. 2). This was initially observed under carbon-limited conditions, such as in fed-batch experiments (Korz et al, 1995; Aiba et al, 1976; Ljunggren and Häggström, 1994) and chemostat experiments (Van Hoek et al, 1998; Paalme et al, 1997; Hayter et al, 1993; Vergara et al, 2018). In carbon-limited chemostats, the relationship between the overflow flux and the dilution rate was first interpreted as a "growth rate dependence" of the overflow flux (Van Hoek et al, 1998; Basan et al, 2015). However, this interpretation is misleading. Because the growth rate is a *consequence*, and not a *cause*, of the imposed restriction on the uptake rate of glycolytic substrates, these results should instead be understood as demonstrating a dependence of the overflow flux on the glycolytic flux rather than on the growth rate (Eiteman and Altman, 2006; Sonnleitner and Käppeli, 1986; Kochanowski et al,

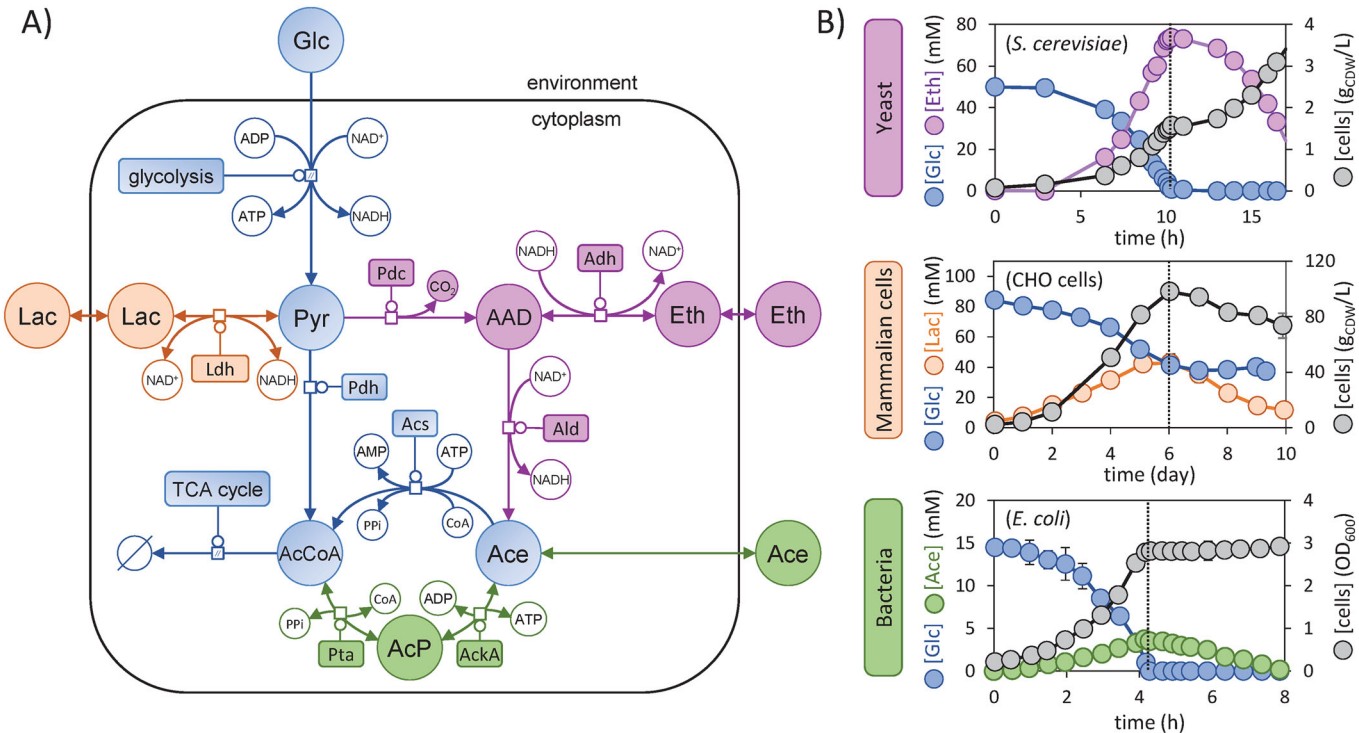

**Figure 1. Overflow metabolism induces diauxic shifts in yeast, bacterial, and mammalian cells.**

(A) Topology of conserved central metabolic reactions (in blue) and of the main overflow pathways in *S. cerevisiae* (in purple), CHO cells (in orange), and *E. coli* (in green). This network follows the Systems Biology Graphical Notation format (http://sbgn.org), with circles representing metabolites and rounded rectangles representing enzymes. (B) Overflow by-products are produced during the glucose consumption phase and reconsumed once glucose is depleted or no longer consumed. The switch is highlighted in each case by a vertical dotted line. Experimental data were taken from ref. (Enjalbert et al, 2013) for *E. coli*, from ref. (Meyenburg, 1969) for *S. cerevisiae*, and from ref. (Mulukutla et al, 2012) for CHO cells. Source data are available online for this figure.

2021). Alternative metabolic engineering strategies targeting the glycolytic flux support this interpretation and confirm the positive control of overflow fluxes by glycolysis when the glycolytic flux reaches an organism-dependent threshold (Millard et al, 2023; Tanner et al, 2018; Elbing et al, 2004). Lowering the activity of enzymes involved in the transport of glycolytic nutrients (phosphotransferase and ABC transporters in *E. coli* (Fuentes et al, 2013) and hexose and maltose transporters in yeast (Elbing et al, 2004) and CHO cells (Wlaschin and Hu, 2007), or in their utilization (e.g., phosphoglucose isomerase and phosphofructokinase in *E. coli* (Long and Antoniewicz, 2019), hexokinase in yeast (Raamsdonk et al, 2001), reduces by-product accumulation. Glycolytic control of overflow fluxes is also apparent upon chemical inhibition of the glycolytic flux (α-methylglucose in *E. coli* (Millard et al, 2023), $H_2O_2$ in yeast (Xiao et al, 2022), 2-deoxyglucose or 5-thioglucose (Naik et al, 2023; Niccoli et al, 2017) in CHO cells), as well as when using glycolytic substrates with a naturally low glycolytic flux, such as glycerol, fructose, or galactose (Altamirano et al, 2006; Gerosa et al, 2015; Ostergaard et al, 1999). In all these experiments, lowering the glycolytic flux was found to limit by-product accumulation.

Overflow fluxes are also influenced by pyruvate and acetyl-CoA utilization pathways, primarily through the TCA cycle. In *E. coli*, overexpressing the global regulator ArcA represses the expression of TCA enzymes, thereby decreasing respiratory capacity and increasing acetate flux (Basan et al, 2017) (Fig. 2). In contrast,

deleting ArcA increases TCA activity and reduces acetate production (Peebo et al, 2014). Deleting *iclR* activates another acetyl-CoA-consuming pathway, the glyoxylate shunt, and also reduces acetate accumulation (Liu et al, 2017). A similar response has been observed for lactate in CHO cells overexpressing *c-myc* (an activator of mitochondrial biogenesis; results for this *cMYC-OE* strain shown in Fig. 2) (Kuystermans et al, 2010; Li et al, 2005) or *MDH2* (Chong et al, 2010). Ethanol production is also higher in "petite" yeast with disrupted mitochondrial activity (results for the *pet191Δ* strain shown in Fig. 2) (Hutter and Oliver, 1998).

In tandem with these experimental approaches, the mathematical framework of metabolic control analysis and the increasing availability of kinetic models of overflow metabolism have contributed to a quantitative understanding of overflow flux regulation (Millard et al, 2021; Mulukutla et al, 2012; Millard et al, 2023). Experimental and modeling results both suggest there is no single "rate-limiting step", contrary to what has often been suggested. Instead, the control of overflow fluxes is widely distributed between three processes revolving around pyruvate: the overflow pathway itself, glycolysis, and the TCA cycle. While the same three pathways have been shown to exert comparable control over overflow fluxes in *E. coli* (Millard et al, 2021), their specific contributions to overflow flux control in yeast and mammalian cells remain to be quantified.

This distributed control pattern, consistently observed across different organisms, may help explain the significant phenotypic

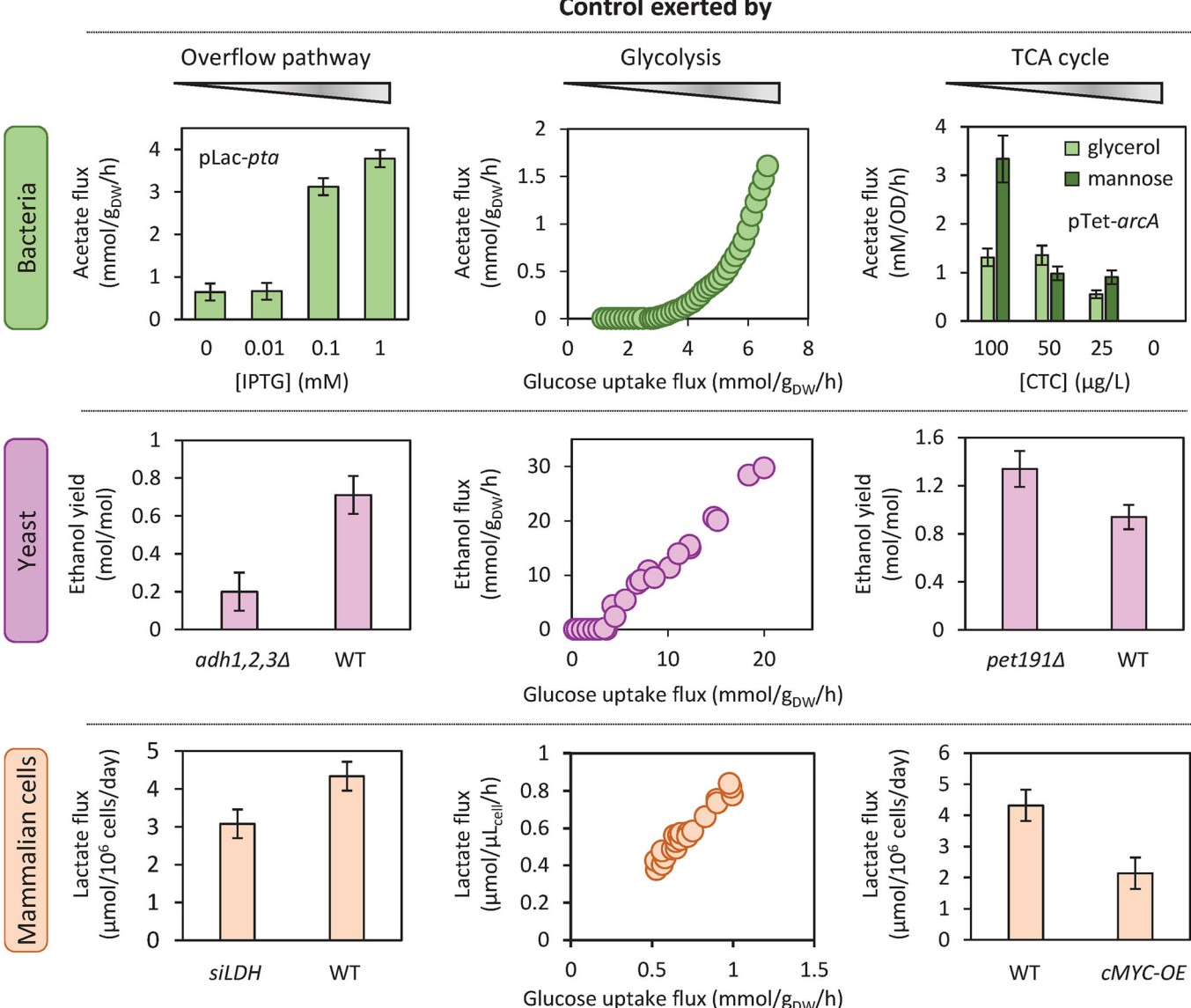

**Figure 2. Identification of the pathways controlling overflow fluxes.**

Overflow fluxes have been measured in response to modulations of the activity of enzymes in the overflow pathway (left column), glycolytic pathway (middle column), and TCA cycle (right column), in bacterial (*E. coli*, top row), yeast (*S. cerevisiae*, middle row), and mammalian cells (CHO cells, bottom row). The overflow flux depends on the activity of all three pathways, demonstrating that they all control overflow fluxes during growth on glycolytic substrates. The *E. coli* data are taken from ref. (Sun-Gu and Liao, 2008) for the overflow pathway, from ref. (Peebo et al, 2014) for glycolysis, and from ref. (Basan et al, 2017) for the TCA cycle. The *S. cerevisiae* data are taken from ref. (Smidt et al, 2012) for the overflow pathway, from refs. (Christen and Sauer, 2011; Van Hoek et al, 1998; Heyland et al, 2009) for glycolysis, and from ref. (Hutter and Oliver, 1998) for the TCA cycle. The CHO cell data are taken from ref. (Noh et al, 2017) for the overflow pathway, from ref. (Tanner et al, 2018) for glycolysis, and from ref. (Kuystermans et al, 2010) for the TCA cycle. Source data are available online for this figure.

diversity of overflow metabolism in different strains in that it is linked to the regulation of glycolysis and the TCA cycle. For instance, flux measurements in seven yeast species have shown that a combination of low glycolytic fluxes with high TCA fluxes leads to higher growth rates and lower ethanol production (Christen and Sauer, 2011). Similarly, variability in overflow metabolism among natural and engineered *E. coli* strains can, at least in part, be attributed to differences in regulation of their glycolytic and TCA cycle activities (Fuentes et al, 2013; Castaño-Cerezo et al, 2015;

Lozano Terol et al, 2019; Waegeman et al, 2011; Marisch et al, 2013; Monk et al, 2016; Baldazzi et al, 2023).

## The high reversibility of overflow metabolism uncouples carbon oxidation from nutrient uptake

While enzymes catalyzing the production of overflow products are reversible in vitro, the flux through these pathways has traditionally been considered irreversible because of the net accumulation of

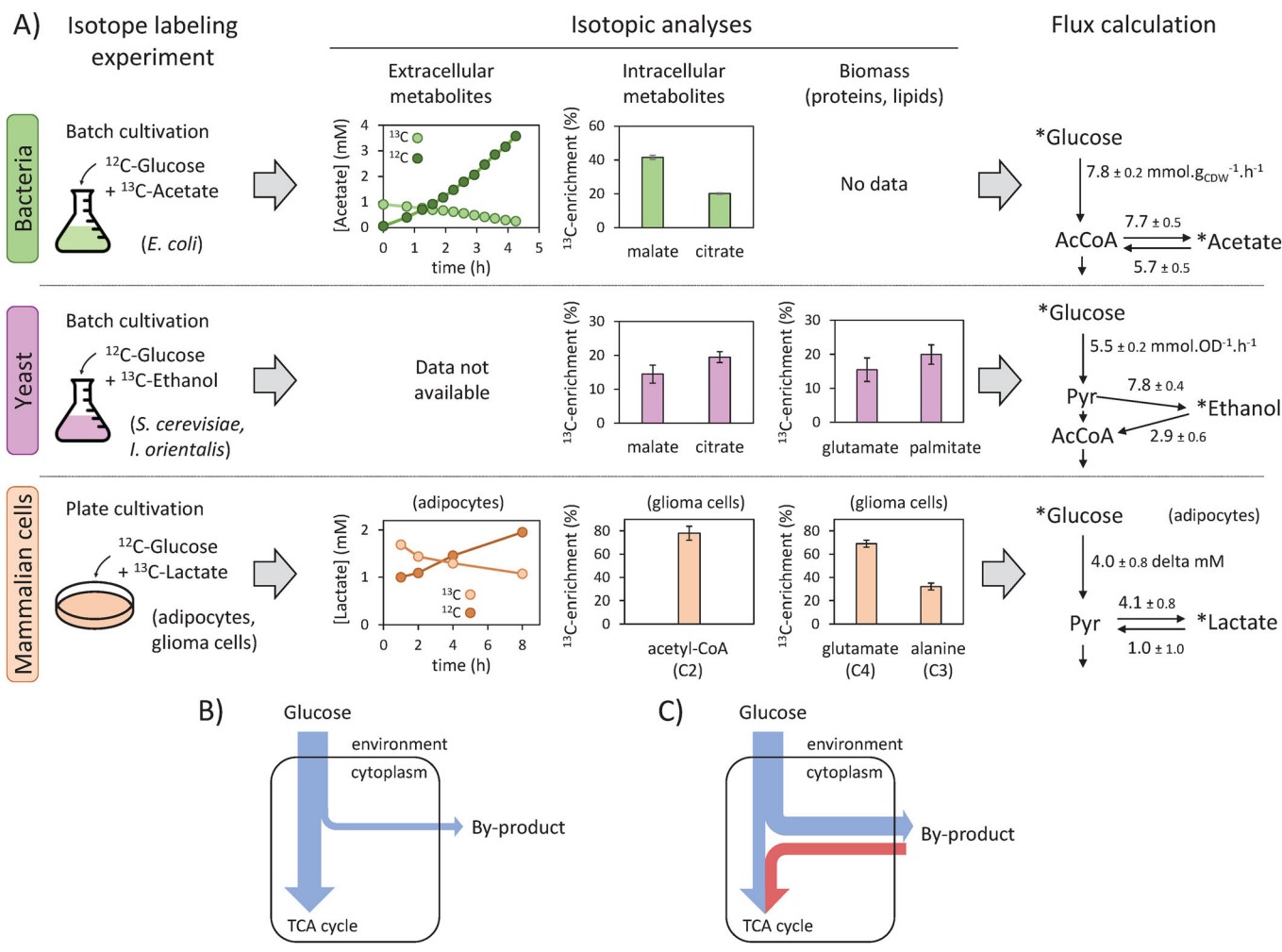

**Figure 3.  Isotope labeling experiments demonstrate the high reversibility of overflow pathways.**

13C-fluxomics allows the quantification of forward and reverse overflow fluxes, revealing the high reversibility of overflow fluxes in bacteria, yeast and mammalian cells (**A**). Isotope labeling experiments were performed by adding small amounts of 13C-by-products during growth on glucose. The concentration dynamics of 12C- and 13C-by-products in the extracellular medium were monitored by NMR. 13C-incorporation was also measured into central metabolites (malate, citrate, and acetyl-CoA are shown as example) and biomass components (glutamate, alanine, and palmitate are shown as example). Isotopic data from *E. coli* and *S. cerevisiae* were measured by MS and are expressed as the mean 13C-enrichment of metabolites, while data from glioma cells were measured by NMR and represent 13C-enrichments at specific carbon atoms within the metabolites (with the carbon atom number indicated below the metabolite name). Fluxes were then quantified using isotopic models. The *E. coli* data are taken from ref. (Enjalbert et al, 2017), the yeast data from ref. (Xiao et al, 2022), and the mammalian data from refs. (Lagarde et al, 2021; Bouzier et al, 1998). Complete datasets are available in the respective publications. In the flux maps, extracellular metabolites are denoted with stars; all other metabolites are intracellular. According to the traditional view of overflow metabolism, a fraction of glucose is excreted as a by-product, with the rest being oxidized through the TCA cycle (**B**). However, recent results indicate that the high reversibility of overflow pathways leads to a partial uncoupling of glucose uptake from glucose oxidation in the TCA cycle (**C**). Source data are available online for this figure.

by-products and the expression of different genes during production and consumption (Wolfe, 2005; Hartley et al, 2018; de Smidt et al, 2008). However, the recent development of 13C-fluxomics approaches, combining isotope labeling experiments with mathematical models of metabolism (Fig. 3A), has revealed that overflow fluxes are also reversible in vivo, even in conditions where catabolite repression is active.

Adding small amounts of 13C-acetate to the extracellular medium during growth of *E. coli* on unlabeled glucose leads to an increase in the concentration of 12C-acetate (produced from glucose) and to a decrease in the concentration of 13C-acetate (Fig. 3A) (Enjalbert et al, 2017). This demonstrates that acetate is constantly exchanged

between *E. coli* cells and their environment and thus that the acetate flux is reversible. Carbon atoms from extracellular 13C-acetate are likewise incorporated into intermediates of the TCA cycle, revealing the role of by-products in carbon nutrition. Similar results have been obtained in yeast (using 13C-ethanol, Fig. 3A) (Xiao et al, 2022), in adipocytes, glioma, and HeLa cells in vitro (using 13C-lactate, Fig. 3A) (Lagarde et al, 2021; Bouzier et al, 1998; Chen et al, 2016), and in vivo in mice (using intravenous infusion of 13C-lactate) (Hui et al, 2020). These isotope labeling experiments showed significant incorporation of 13C-by-products into TCA intermediates, amino acids and fatty acids in *E. coli*, yeast and mammalian cells (Fig. 3A). Furthermore, growth of yeast or mammalian cells on unlabeled glucose plus

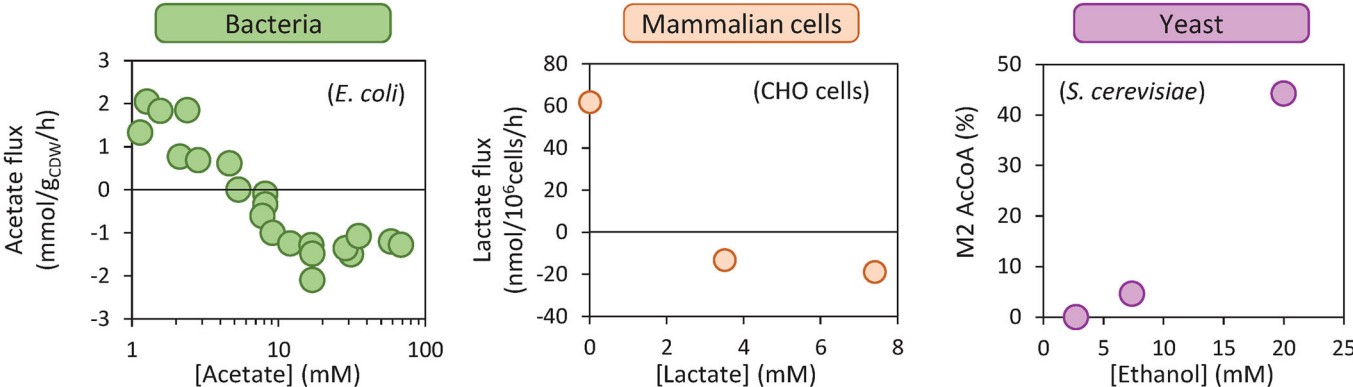

**Figure 4. Overflow fluxes are thermodynamically controlled by the concentration of overflow metabolites.**

In *E. coli* grown on glucose, the acetate flux decreases when the extracellular acetate concentration is increased (Enjalbert et al, 2017). In CHO cells grown on galactose, increasing the extracellular lactate concentration reverses the lactate flux (Altamirano et al, 2000; Torres et al, 2019). In *S. cerevisiae* grown on glucose, the contribution of [13]C-ethanol to acetyl-CoA biosynthesis increases with the extracellular ethanol concentration (Xiao et al, 2022). Although the net ethanol flux was not measured in these experiments, these results imply an increase in ethanol uptake. Source data are available online for this figure.

[2]H-ethanol or [2]H-lactate led to the incorporation of [2]H isotopes into NAD(P)H, confirming the high reversibility of overflow pathways and revealing the contribution of overflow metabolites to redox metabolism (Hui et al, 2020; Xiao et al, 2022). [13]C-labeling experiments in *E. coli* mutant strains missing different acetate pathway enzymes have shown that those responsible for acetate production (Pta and AckA) are also simultaneously involved in acetate consumption (Enjalbert et al, 2017). Similarly, Ldh contributes to both lactate production and utilization in adipocytes (Lagarde et al, 2021). The contributions of different overflow pathway enzymes to bidirectionality have not been investigated in detail in yeast and CHO cells, but the producing enzymes may also be (at least partly) responsible for the simultaneous utilization of by-products.

Integration of isotopic data collected in [13]C- or [2]H-labeling experiments into isotopic models has enabled the quantification of individual production and consumption fluxes. While the exact flux values vary depending on the model and underlying assumptions, all *E. coli*, yeast and mammalian cell models converge on an exchange flux in the same range as glucose uptake and 2–4 times higher than the net overflow flux (Fig. 3A) (Enjalbert et al, 2017; Lagarde et al, 2021; Xiao et al, 2022). In these organisms therefore, the high reversibility of overflow pathways partly uncouples glucose utilization from glucose oxidation: a large fraction of glucose (between 33 and 51% of carbon, Fig. 3A) is catabolized into extracellular overflow products, which are simultaneously taken up and used through the TCA cycle as carbohydrate sources (Fig. 3B, C). This partial uncoupling of glycolysis from the TCA cycle via overflow products is therefore a conserved feature of prokaryotic and eukaryotic metabolism.

## Thermodynamic control of overflow fluxes can lead to co-consumption of overflow metabolites with glycolytic nutrients

The recent discovery that overflow is a highly reversible process has led to the suggestion that increasing the extracellular concentration of by-products could modulate the thermodynamic gradient of overflow pathways and, through this thermodynamic driving force, regulate their flux. This hypothesis of local thermodynamic control of overflow pathways has been tested in detail in *E. coli* (Enjalbert et al, 2017). In line with the predictions obtained from a thermodynamic model of the Pta-AckA pathway, experimental data confirmed that increasing the acetate concentration progressively decreases the net acetate production flux (Enjalbert et al, 2017), abolishing acetate accumulation at ~10 mM (Fig. 4). Beyond this threshold, the net acetate flux reverses, leading to the co-utilization of acetate with glucose and other glycolytic substrates (fucose, gluconate, glycerol, galactose) (Enjalbert et al, 2017; Millard et al, 2023). Analysis of *E. coli* mutant strains missing different enzymes in the acetate pathway has shown that co-utilization of acetate with glycolytic nutrients is supported solely by the Pta-AckA pathway (Millard et al, 2023; Enjalbert et al, 2017).

Thermodynamic control of overflow fluxes in yeast and CHO cells has yet to be studied in detail, but we predict that similar principles apply to these organisms. This hypothesis is supported by the following observations. CHO cells have been observed to consume lactate alongside glycolytic nutrients (glucose or galactose) (Altamirano et al, 2006; Martínez-Monge et al, 2019), and increasing the extracellular lactate concentration during growth on galactose has been shown to reverse the lactate flux (Altamirano et al, 2000; Torres et al, 2019) (Fig. 4). In yeast grown on glucose, increasing the ethanol concentration enhances the incorporation of extracellular [13]C-ethanol into acetyl-CoA (Xiao et al, 2022) (Fig. 4), indicating an increase in extracellular ethanol uptake. The net ethanol flux was not measured in these experiments, but co-consumption of glucose and ethanol has already been observed (Raamsdonk et al, 2001). These data support our hypothesis that overflow fluxes in all organisms are largely controlled at the thermodynamic level by the concentration of the respective by-products (relative to the concentration of internal metabolites), as observed for other nutrients in mammalian cells (Li et al, 2022a). As demonstrated in detail for *E. coli* (Enjalbert et al, 2017) and suggested from published data for yeast and CHO cells, increasing the concentration of overflow metabolites does not only reduce

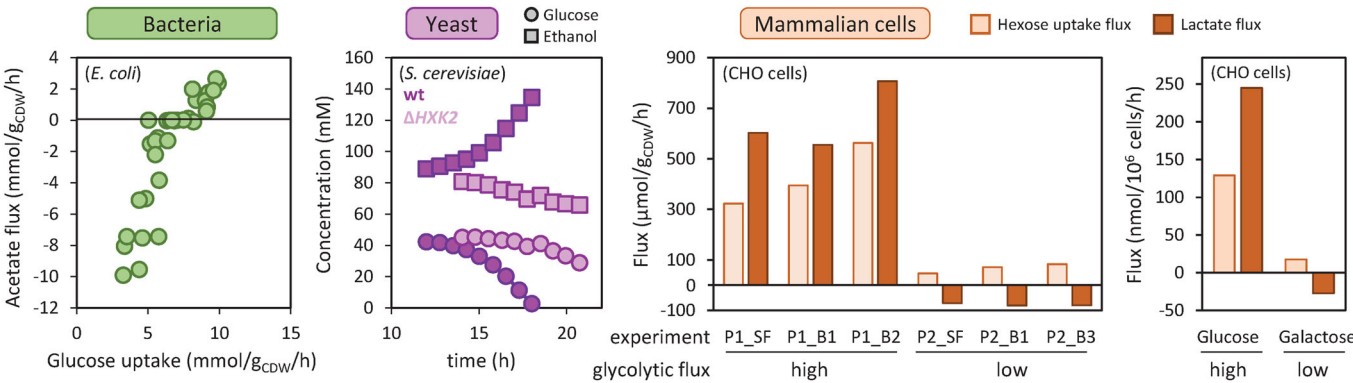

**Figure 5.  Impact of reducing the glycolytic flux on overflow fluxes.**

In *E. coli*, *S. cerevisiae*, and CHO cells, lowering the glycolytic flux (either by increasing the concentration of the glycolytic inhibitor α-methylglucose in *E. coli* (Millard et al, 2023), by deleting *HXK2* in *S. cerevisiae* (Raamsdonk et al, 2001), or by using different bioprocess conditions or carbon sources in CHO cells (Altamirano et al, 2006; Martínez-Monge et al, 2019)) reverses the overflow flux. The overflow flux switches from production (positive values) at high glycolytic flux to co-utilization (negative values) when the glycolytic flux is low. Source data are available online for this figure.

their accumulation but may also lead to the reversal of overflow fluxes. The role of overflow metabolites as by-products or co-substrates of glycolytic nutrients is thus determined by this thermodynamic control mechanism.

## Low uptake of glycolytic nutrients enhances co-utilization of overflow metabolites

Studies of glycolytic control of overflow fluxes have mostly been conducted in the absence of by-products. However, flux control is a local property that depends on the metabolic state of the organism, influenced in turn both by the glycolytic flux and the concentration of by-products. The intricate interplay between glycolytic and overflow fluxes in the presence of acetate has been explored in detail in *E. coli* (Enjalbert et al, 2017; Millard et al, 2021, 2023). A kinetic model predicted that decreasing the glycolytic flux would reduce the acetate flux in the absence of acetate, and that the same perturbation would enhance acetate utilization in the presence of acetate. This negative control of the acetate flux by glycolysis in the presence of acetate was experimentally confirmed by lowering the glycolytic flux using different approaches (Millard et al, 2023) (Fig. 5). The glycolytic flux was found to control both the acetate flux and the acetate concentration at which the acetate flux reverses, this threshold being lower at low glycolytic flux (Millard et al, 2023). This non-monotonic control relationship and its implications have not been explored in yeast and CHO cells, but here again several observations suggest that it likely applies to these organisms as well. Co-consumption of lactate and glucose has indeed been observed in CHO cells, but only when the glucose uptake flux is low (Torres et al, 2019; Ahn and Antoniewicz, 2011) (Fig. 5). Growing CHO cells on galactose, a nutrient with a naturally low glycolytic flux, also leads to co-utilization of lactate (Altamirano et al, 2006). In yeast, simultaneous consumption of glucose and ethanol has been observed at low glycolytic flux in a *S. cerevisiae* strain lacking *HXK2* (Raamsdonk et al, 2001), the gene encoding the main glucose transporter (Fig. 5). Moreover, exposure of yeast to $H_2O_2$ also leads to a decrease in glycolytic flux (Shenton and Grant, 2003) and increases the contribution of ethanol to NADH regeneration

(Xiao et al, 2022). These observations are consistent with the relationship identified in *E. coli*, suggesting that a similar control mechanism exists in yeast and CHO cells, though further work is required to test this hypothesis in detail.

## Overflow metabolites are global regulators of cellular metabolism and physiology

Overflow metabolites fundamentally alter cellular physiology. Lactate, for example, is known to modulate growth, immunity, tumorigenesis, and tumor development (Brooks, 2018). Acetate influences growth, virulence, motility, carbon metabolism, biofilm formation, and peptide uptake (Castaño-Cerezo et al, 2014; Millard et al, 2021). Ethanol triggers a global response in yeast, impacting growth, cell viability, metabolism, cell structure, and membrane function (Sahana et al, 2024). Extensive research has been conducted to uncover the underlying mechanisms. Acetate and lactate can disrupt the intracellular pH and the proton motive force through their uncoupling effect (Pinhal et al, 2019; Brooks, 2018). Ethanol disrupts yeast membrane bilayers, reduces water availability, affects enzymatic activity (Sahana et al, 2024), and denatures proteins. Lactate likewise alters protein stability (Li et al, 2022b). In addition to these physicochemical effects, cells respond to overflow metabolites by reorganizing at all molecular levels, including the transcriptome (Millard et al, 2021, 2023; Barbieri et al, 2023; Sahana et al, 2024), proteome (Kirkpatrick et al, 2001; Sahana et al, 2024; Zhang et al, 2019), and metabolome and fluxome (Millard et al, 2021, 2023). The thermodynamic control of overflow pathways enables direct, system-wide control of fluxes and metabolite concentrations by the by-products themselves (Millard et al, 2021). In addition to this direct control, overflow metabolites also modulate the expression of hundreds of genes involved in most cellular functions (metabolism, replication, motility, etc) (Castaño-Cerezo et al, 2014; Brooks, 2018; Sahana et al, 2024). Though a complete picture is still lacking, some sensing and signaling pathways have been identified. For instance, lactate is an agonist of the G-protein-coupled receptor GPR81 in neurons (Li et al, 2022b), and acetate can be sensed by *E. coli* via the BarA/UvrY

two-component system (Alvarez et al, 2021). Lactate and acetate also modulate the activity of many proteins through epigenetic and post-translational modifications, respectively lactylation and acetylation (Li et al, 2022b; Schastnaya et al, 2023). Several acetylation and lactylation targets involved in key cellular functions such as replication and metabolism have been identified (Xu et al, 2024; Wolfe, 2016), but here again a more global analysis is required. Despite extensive research and continuous progress on this topic, the complete list of targets and the detailed regulatory networks that sense and respond to overflow metabolites remain to be established.

Overflow metabolites are often referred to as "stress factors". Recognizing these compounds as global regulators is crucial to characterize the corresponding regulatory programs and the induced cellular responses. Importantly, the presence of these compounds in the extracellular medium means they can act as messengers between cells, and between organs in mammals, thereby influencing entire cell communities (Baker and Rutter, 2023).

## Overflow metabolites are not toxic per se and can be beneficial to cells

Overflow products have long been considered toxic waste because of their detrimental effect on growth. Different theories have been proposed to explain this growth inhibition, but most cannot be generalized to all types of by-products (e.g., alcohols, acids, ketones). Acetate (Luli and Strohl, 1990), ethanol (Brown et al, 1981) and lactate (Lao and Toth, 1997) all hinder cellular membrane function either through their decoupling effect (Brooks, 2018; Pinhal et al, 2019), deleterious change in membrane fluidity, or osmotic stress. These by-products can also interfere with intracellular amino-acid pools such as methionine, glutamate, and tryptophan (Pinhal et al, 2019). Ethanol also promotes protein denaturation in yeast (Sahana et al, 2024). These long-known deleterious effects have contributed to the established perception of overflow metabolites as toxic compounds.

However, recent studies paint a very different picture, in which these molecules may in fact be beneficial for their producers. Lactate has recently been shown to be continuously used by mammalian cells, even under conditions of net lactate production, with many positive properties, such as (i) being a source of pyruvate and energy (Hui et al, 2020), (ii) being an energy redistribution shuttle between cells (Brooks, 2018; Rabinowitz and Enerbäck, 2020), (iii) improving viability (Mulukutla et al, 2012), and (iv) acting as an antioxidant (Groussard et al, 2000). In yeast, ethanol participates in carbon nutrition and is a major source of NADH and NADPH (Xiao et al, 2022), with its contribution augmented under oxidative stress conditions. Acetaldehyde, an intermediate of ethanol metabolism described as detrimental for growth, may instead promote growth and reduce lag in S. cerevisiae (Walker-Caprioglio and Parks, 1987), even at high concentrations (up to 140 mM). In bacteria, acetate has recently been shown to increase the robustness of E. coli to glycolytic perturbations by buffering carbon uptake (Millard et al, 2023). Acetate even boosts E. coli growth when the glycolytic flux is reduced, clearly demonstrating that it is not always toxic and questioning previous mechanistic explanations for its inhibitory effect on growth (Millard et al, 2023).

Although our degree of understanding of the impact of by-products varies between the three types of organisms, the parallels are striking. Overflow by-products should cease to be viewed solely as toxic waste and instead be recognized in some situations as beneficial co-substrates for individual cells and cell populations.

## Similar principles apply to other species and overflow by-products

We have focused on the most abundant and best studied overflow products in a selection of model species, but overflow is a common feature of living organisms, including bacterial, mammalian, fungal, and plant cells. It can also arise at virtually any step of the metabolic network, depending on the strain, cultivation conditions, and experimental setup (e.g., carbon source, medium composition, temperature, pH, continuous or batch cultivation). It is therefore no surprise that overflow phenomena have been reported for a wide range of compounds, including sugars, amino acids, organic acids, and nucleotides (Reaves et al, 2013; Ser et al, 2016; Phégnon et al, 2024; Paczia et al, 2012). To list just a few examples, glucose overflow has been observed in engineered S. cerevisiae and E. coli strains grown on xylose (Diaz et al, 2019; Nijland et al, 2021), overflow of arabitol has been observed in Pichia pastoris grown on glycerol (Fina et al, 2021), and hemiterpenoid glycosides and acetate overflows have been observed in plants under stress conditions (Ward et al, 2011). Overflow has been observed in many species for most central metabolites, though at lower concentrations (Paczia et al, 2012), and does not necessarily involve carbon compounds, with ammonium overflow having been observed during the growth of E. coli on mixtures of glycerol and ethanolamine (Jallet et al, 2024).

Several clues suggest that the principles discussed in this review may be generic and apply equally to most species and by-products. First, as observed for central metabolites and ammonia (Paczia et al, 2012; Jallet et al, 2024), overflow metabolites are typically reconsumed following the depletion of the primary nutrient. Second, the control of overflow fluxes is often shared between producing and consuming pathways, as shown for pyrimidine overflow in E. coli and poly-γ-glutamic acid overflow in Bacillus licheniformis (Li et al, 2021; Reaves et al, 2013), thereby reflecting the imbalance between the two pathways. Overflow typically occurs under conditions of nutrient excess (Paczia et al, 2012), characterized by high biosynthetic fluxes that cannot be supported by downstream pathways, while nutrient limitation favors the co-utilization of overflow metabolites (Paczia et al, 2012; Ward et al, 2011). Overflow also occurs when the uptake rates of different nutrients are imbalanced, with one nutrient limiting the downstream pathways required to incorporate the other. This has been observed in E. coli grown on a mixture of glycerol (used as a carbon source) and ethanolamine (used as a nitrogen source) (Jallet et al, 2024). Finally, overflow fluxes obey thermodynamic control, with high extracellular concentrations of metabolites limiting production and favoring (co-)utilization (Paczia et al, 2012; Pastoors et al, 2023). One of the most striking examples of the latter is perhaps the reversal of the TCA cycle in Hippea maritima at high $CO_2$ levels, enabling $CO_2$ fixation (Steffens et al, 2021).

## Toward a unified theory of overflow metabolism

While this analysis of the literature reveals the similarities in overflow metabolism across multiple organisms, these shared

principles do not explain why overflow occurs. We now examine the prevailing theories of overflow metabolism, focusing on those that potentially apply to all living organisms, in light of the principles outlined above.

Cells operate under multiple universal biophysical constraints, some of which may drive overflow metabolism. These constraints may arise from resource allocation (Basan et al, 2015; Shen et al, 2024; Vazquez and Oltvai, 2016), intracellular crowding (Vazquez and Oltvai, 2016; Vazquez et al, 2010), cell geometry (Zhuang et al, 2011; Szenk et al, 2017), or limits on energy dissipation (Niebel et al, 2019; Saadat et al, 2020). Although these theories are based on different aspects of cell physiology, they make the same prediction that overflow metabolites are only produced at high glucose uptake rates. Therefore, they provide independent rationales for the emergence of overflow.

One particularly compelling set of theories links overflow metabolism to spatial constraints. Zhuang et al propose that acetate overflow in *E. coli* results from competition for membrane space between glucose transporters and respiratory chain components. ATP production by the cytosolic overflow pathway requires less membrane space than respiration, forcing cells to favor overflow when membrane space is limiting (Zhuang et al, 2011). Similarly, Szenk et al hypothesize that when the bacterial membrane becomes saturated, cells shift from membrane-bound respiratory enzymes to cytosolic overflow pathways to maintain optimal energy production (Szenk et al, 2017). Other studies suggest that compartment-specific macromolecular crowding also imposes a fundamental limit on how much oxidative metabolism a cell can sustain, potentially driving overflow in bacterial, yeast and mammalian cells (Vazquez and Oltvai, 2016; Vazquez et al, 2010; Elsemman et al, 2022). Together, these theories position overflow metabolism as a necessary adaptation to maximize growth under spatial constraints, though the regulatory mechanisms underlying this shift remain to be identified.

Overflow metabolism may also be driven by metabolic economics. Basan et al, argue that bacterial oxidative metabolism demands substantial enzyme investment to produce ATP, whereas overflow metabolism, though less efficient for ATP production, requires fewer enzymes (Basan et al, 2015). This trade-off between energy production efficiency and enzyme cost could explain the emergence of overflow in *E. coli* (Basan et al, 2015). Recently, Kukurugya et al, suggested that this proteome allocation model may also apply to yeast and mammalian cells (Kukurugya et al, 2024), although this remains a topic of active debate (Shen et al, 2024).

Another theory is that overflow metabolism is driven by global thermodynamic constraints. Niebel et al, propose that cells operate within a finite Gibbs energy dissipation rate, meaning that as substrate uptake rates increase, cells must eventually shift from respiration (high dissipation rate) to overflow pathways (low dissipation rate) once their energy dissipation threshold is reached (Niebel et al, 2019; Saadat et al, 2020). This theory successfully explains the production of overflow metabolites in *S. cerevisiae* and *E. coli* cells, but its relevance to mammalian cells remains untested. Furthermore, the biophysical basis of this energy dissipation limit remains to be clarified (Niebel et al, 2019; Saadat et al, 2020; Losa et al, 2022; Yang et al, 2021).

All of the aforementioned theories assume that cells have evolved toward optimal growth, a strong and still controversial

assumption (Shen et al, 2024; Schuetz et al, 2012; Towbin et al, 2017; Basan et al, 2020; Mori et al, 2019). Alternative theories propose that rather than simply being a consequence of growth optimization under biophysical constraints, overflow metabolism is itself an advantageous adaptation. For instance, ethanol overflow in yeast may be an adaptation to fluctuations in oxygen availability (Shen et al, 2024), while acetate overflow in *E. coli* may increase the number of offspring cells in nutrient-fluctuating environments (Rabbers et al, 2022). Other studies have proposed that overflow confers a competitive advantage by enabling cells to rapidly consume shared resources before competing species do (Rozpędowska et al, 2011) or by limiting the buildup of toxic reactive oxygen species (Brand, 1997).

None of these theories has been accepted as a universal explanation for overflow, neither do they explicitly incorporate the functioning principles highlighted in this review. In particular, none of these models were developed with the reversibility of overflow pathways in mind, nor the simultaneous utilization of overflow metabolites with glycolytic substrates or the potential benefits of overflow metabolites to the producing cells. Whether these theories can fully accommodate the outlined principles is therefore questionable and should be investigated.

An alternative theory developed in *E. coli* is that overflow metabolism is largely governed by thermodynamic gradients between cells and their environment (Enjalbert et al, 2017). This theory accurately predicts acetate secretion in *E. coli* in the absence of extracellular acetate and the fact that increasing acetate concentrations can suppress or even reverse the acetate flux. Interestingly, the data presented in this study suggest that this theory may be applicable to yeast and mammalian cells, though further work is required to test this hypothesis in detail. While local thermodynamic forces are key drivers of flux reversal and co-utilization of overflow metabolites, enzymatic regulation is equally critical, indicating that overflow metabolism operates in a multi-layered regulatory system. Indeed, a kinetic model accounting for enzyme saturation and thermodynamic gradients fails to predict observed phenotypes in *E. coli* unless acetate-driven modulation of glycolytic and TCA cycle activities is also included (Millard et al, 2021). This model explains growth inhibition by acetate at high glycolytic flux, while also accurately predicting its beneficial effect at low glycolytic flux (Millard et al, 2023, 2021) and is, to our knowledge, the only existing model to integrate both the regulation of the overflow pathway and the cellular response to overflow metabolites, two aspects that have historically been treated as separate phenomena. Key predictions of this theory have been validated in *E. coli* and the proposed regulatory interplay has been supported by transcriptomics data (Millard et al, 2023, 2021), though the underlying molecular regulatory network remains to be identified. Moreover, while thermodynamic gradients and enzymatic regulation explain why overflow occurs and offer insight into the advantage of an active overflow pathway under fluctuating nutritional environments (Enjalbert et al, 2017; Millard et al, 2023), the physiological relevance of the observed phenotypes should be further explored.

Ultimately, no single theory has yet been conclusively validated as a universal explanation for overflow metabolism. It is unclear how the different theories interrelate or contradict each other, even within a single organism. In an effort to compare existing theories, de Groot et al, reformulated several constraint-based models into a

standardized mathematical framework and found that they all predict overflow metabolism whenever two growth-limiting constraints are present, although the nature of these constraints remains unspecified (de Groot et al, 2020). Determining whether current theories are mutually exclusive, complementary, or dependent on the organism or environmental conditions will require further studies. Clarifying how these theories align with the outlined principles will also be essential to challenge, refine, and integrate current hypotheses. For instance, the fact that supplementation with just a few millimolar acetate suppresses acetate production by *E. coli* (Enjalbert et al, 2017; Millard et al, 2021, 2023), without significantly affecting growth, challenges theories asserting that rapid growth is inherently dependent on the production of overflow metabolites. Similarly, the positive effect of overflow metabolites on growth in certain conditions undermines theories that predict only negative effects. Testing current theories under conditions where overflow metabolites are co-utilized with glycolytic substrates also presents an opportunity to uncover new metabolic principles.

A comprehensive re-evaluation and integration of the proposed theories will be essential to uncover the fundamental causes of overflow metabolism, and its role and regulation. The principles highlighted in this study can serve as criteria for evaluating existing and future theories, providing a set of observations that any robust explanation should account for. Only by bridging these gaps can we move toward a potential unified theory explaining this widespread phenomenon across all domains of life.

## Conclusions

Despite extensive data on the universally observed phenomenon of overflow, no generic framework has yet been proposed to explain both the determinants of overflow fluxes and the response of producing cells to the resulting by-products. This review offers a new perspective, emerging from the integration and reinterpretation of various observations from several fields in different organisms, which points to the existence of conserved principles in prokaryotes and eukaryotes (Fig. 6): (i) overflow metabolism is a reversible process that partially uncouples carbon oxidation from nutrient uptake, (ii) overflow fluxes are regulated at the thermodynamic and metabolic levels, with metabolic control being distributed between several pathways, (iii) overflow metabolites can act as co-substrates for glycolytic nutrients, with (iv) their role as nutrient or by-product determined by their concentration and the uptake rate of other nutrients, (v) reducing the uptake of glycolytic nutrients enhances the co-utilization of overflow metabolites, and (vi) overflow metabolites should not always be considered stress factors but global regulators of metabolism with (vii) several dedicated regulatory programs and (viii) many positive roles for cells and populations. The proposed conceptualization explains and unifies previous observations, such as the co-utilization of ethanol and glucose by the *hxk2Δ* yeast mutant, the augmented contribution of ethanol to yeast metabolism in response to oxidative stress or when ethanol concentration is increased, and the reversal of lactate flux in CHO cells between different growth phases on glucose. It also sheds new light on overflow, traditionally viewed as a unidirectional process leading to the production and accumulation of compounds that are toxic

to producing cells, whereas in fact the high reversibility of overflow fluxes enables their utilization as nutrients. This suggests that the production and the co-utilization of overflow metabolites are two faces of the same coin. However, studies on the co-utilization of different carbon sources (e.g., Hermsen et al, 2015; Wang et al, 2019; Harder and Dijkhuizen, 1982; Okano et al, 2019) typically treat overflow metabolism as a distinct phenomenon. Given their potential interconnection, future research should explore whether current theories of nutrient co-utilization could also apply to overflow metabolites. Overflow metabolites have several beneficial effects on carbon and energy metabolism, ultimately improving robustness to a range of environmental perturbations (such as nutrient limitation or oxidative stress). These exchanged metabolites should be considered valuable nutrients and messengers, allowing cells to shape their environments and share resources and information.

Although some of these principles have recently been established in a variety of species (in particular, the reversibility of overflow fluxes), research is typically siloed to individual organisms. The proposed framework calls for dedicated studies to determine whether findings in certain species apply universally, as suggested here by the combined interpretation of results from a heterogeneous corpus. Future studies may benefit from recent methodological advances, particularly from the combination of mathematical models with isotope labeling experiments which allow quantitative, system-level characterization of overflow processes. For instance, understanding the dynamics of SCFAs in the gut could leverage approaches used to clarify the dynamics of circulating lactate. Likewise, methods and models developed to characterize acetate overflow in bacteria could readily be applied to investigate ethanol overflow in yeasts and lactate overflow in mammalian cells. These cross-species applications will refine the proposed principles and likely uncover new ones. Moreover, overflow metabolism is likely governed by a combination of thermodynamic, enzymatic, biophysical, and evolutionary constraints. Future research should focus on integrating existing theories and testing them in light of the outlined principles. These principles, which are basically empirical generalization from observations of overflow metabolism, can therefore serve as criteria for evaluating existing (and future) theories, as a set of observations that any theory should be able to explain. The regulatory programs triggered by overflow metabolites will also need to be identified. Some overarching questions, such as the link between overflow metabolism and cellular heterogeneity, are only just beginning to be explored in certain organisms (Lin and Jacobs-Wagner, 2022).

In biotechnology, the proposed principles can inform the development of more efficient strains, consortia and bioprocesses to maximize overflow fluxes for high-value products and minimize (or even reverse) fluxes for undesired compounds. Considering the nonlinear effects of by-products on growth is also essential for optimizing productivity. These principles could thus lead to more efficient and sustainable bioprocesses, mitigating the negative effects of by-products and enabling their efficient use as co-substrates. In healthcare, this understanding could help elucidate the roles and dynamics of overflow metabolites exchanged within the microbiota, between organs, and between the host and its microbiota, with applications in diagnosis and therapy. Studies of individual cells may be helpful in this context, because they facilitate the identification of cellular processes involved in their

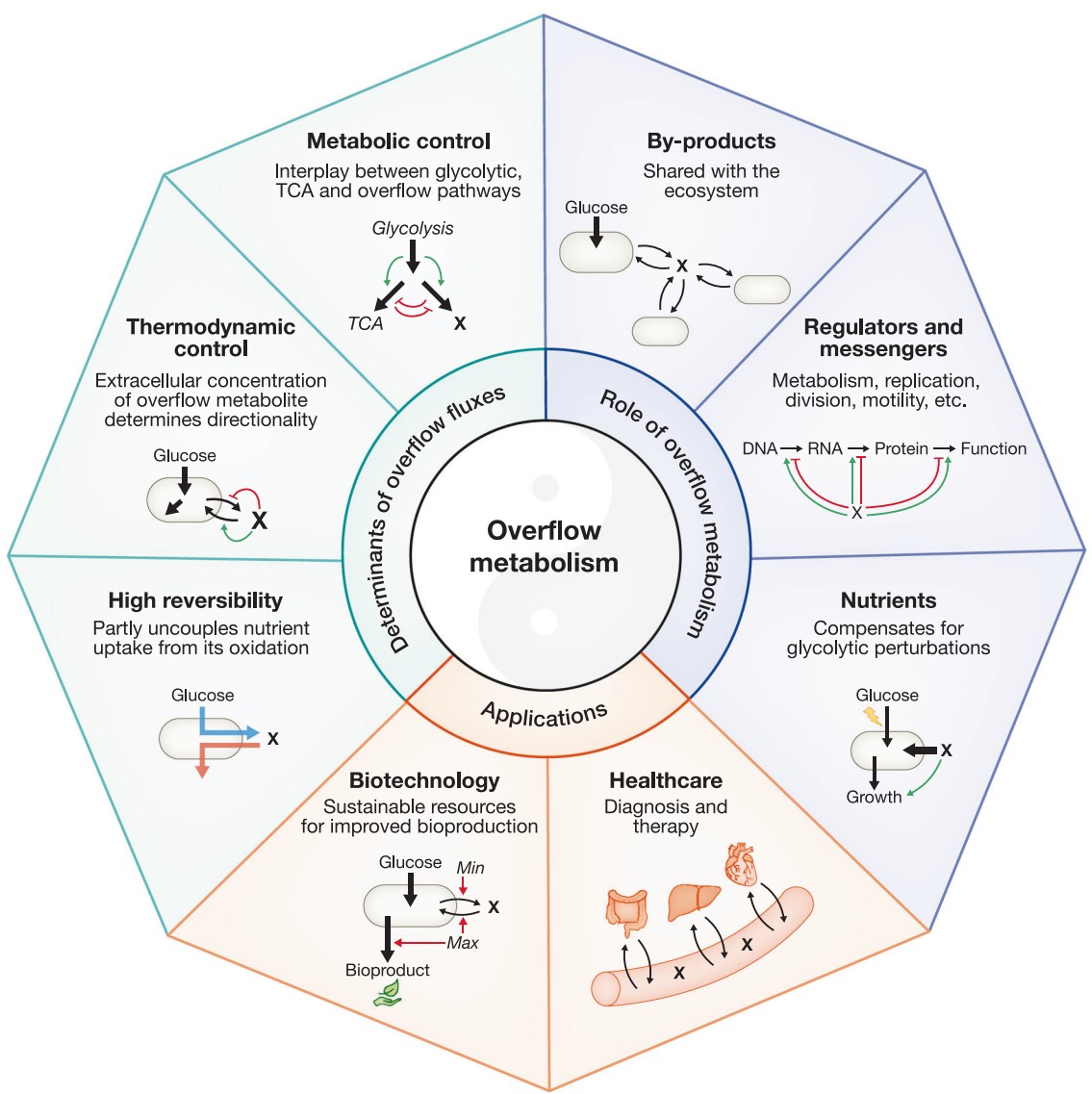

**Figure 6.**   Function and regulation of overflow metabolism (in green), roles of overflow metabolites (in blue) and applications (in orange).

dynamics and of the underlying principles. Integrating this information into combined mathematical models of host and microbiota will likely lead to a quantitative, system-level understanding of their overall dynamics, as developed in recent years for lactate (Li et al, 2022b; Rabinowitz and Enerbäck, 2020). Developing a generic conceptual framework for overflow metabolism—as universal as possible, as specific as necessary—is therefore crucial to advance both fundamental and applied science.

## Peer review information

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

## Acknowledgements

The PhD thesis of Thomas Gosselin-Monplaisir was funded by the Région Occitanie and the MICA department of INRAE (grant COCA-COLI). This work was also supported by the Agence Nationale de la Recherche under the Investissements d'Avenir Programme (grant ANR-18-EURE-0021). The authors thank Benjamin Pfeuty (Université de Lille, France), Sara Castaño Cerezo (Toulouse Biotechnology Institute, France), Guy Lippens (Toulouse Biotechnology Institute, France), and Stéphane Guillouet (Toulouse Biotechnology Institute, France) for insightful discussions on the topic.

## Author contributions

**Thomas Gosselin-Monplaisir**: Conceptualization; Investigation; Visualization; Writing—original draft; Writing—review and editing. **Brice Enjalbert**: Investigation; Writing—original draft; Writing—review and editing. **Sandrine Uttenweiler-Joseph**: Investigation; Writing—review and editing. **Jean-Charles Portais**: Investigation; Writing—review and editing. **Stéphanie Heux**: Investigation; Writing—original draft; Writing—review and editing. **Pierre Millard**: Conceptualization; Supervision; Funding acquisition; Investigation; Visualization; Writing—original draft; Project administration; Writing—review and editing.

## Disclosure and competing interests statement

The authors declare no competing interests.

