## [Peer Review File · Molecular Systems Biology]

Overflow metabolism in bacterial, yeast, and mammalian cells: different names, same game

Thomas Gosselin-Monplaisir, Brice Enjalbert, Sandrine Uttenweiler-Joseph, Jean-Charles Portais, Stéphanie Heux, and Pierre Millard

Corresponding author(s): Pierre Millard (millard@insa-toulouse.fr)

Review Timeline:

Submission Date:	2nd Dec 24
Editorial Decision:	12th Mar 25
Revision Received:	23rd Apr 25
Editorial Decision:	3rd Jul 25
Revision Received:	11th Aug 25
Accepted:	22nd Aug 25

Editor: Poonam Bheda

Transaction Report:

12th Mar 2025

Manuscript Number: MSB-2024-12779

Title: Overflow metabolism in bacterial, yeast, and mammalian cells: different names, same game

Dear Dr Millard,

Thank you again for submitting your work to Molecular Systems Biology. We have now heard back from the two reviewers who agreed to evaluate your review. As you will see below, the reviewers appreciate that the topic is of general interest and timely. However, they raise a series of concerns, which we would ask you to address in a major revision. Specifically, the revision should be focused on better discussing explanatory frameworks proposed for overflow metabolism, which according to both reviewers is currently missing major previous works and/or descriptive. In general it will also be important to tone down statements, e.g. how overflow uncouples glycolysis from TCA cycle and the lack of an overarching framework of overflow.

On a specific note, in a cross-commenting session between the reviewers and the editor, Reviewer 2 also found the two final sections on biotech and health rather superficial with many examples but few original insights, and agreed that these sections could either be removed to make additional space for the deeper literature review and analysis in previous sections, or could be combined and shortened into a final brief concluding section. Finally, Reviewer 2 agreed with Reviewer 1 that the review would require major rewriting to be useful for the community and suitable for publication.

If you feel you can satisfactorily address these points and those listed by the referees, you may wish to submit a revised version of your manuscript. Please include a point-by-point response giving details of the way in which you have handled each of the points raised by the referees. A revised manuscript will once again be subject to review and we cannot guarantee at this stage that the eventual outcome will be favorable. If you would like to discuss further the points raised by the referees, I am available to do so via email or video. Let me know if you are interested in this option.

I look forward to receiving your revised manuscript.

Yours sincerely,

Poonam Bheda, PhD
Scientific Editor
Molecular Systems Biology

Reviewer #1:

The manuscript by Gosselin-Monplaisir reviews the phenomenon of overflow metabolism, drawing upon results from bacteria, yeast, and mammals. The corresponding literatures, starting more than a century ago, have been often read in a disconnected way and the ambition of this review is to identify features that are generic across overflow metabolism in all of these different organisms. Maybe the most interesting novelty of the perspective that is proposed, supporting the claim of providing "fundamental insights" and "universal principles" (abstract), is the idea that overflow metabolites are not to be considered toxic waste, but nutrients and regulators of both cells generating the overflow and cells of other species in a microbial community.

The review is interesting, well-informed, and well-written, but can be improved and/or clarified on a number of points.

Major comments

1. Fig. 3C summarizes one of the main points of this review, namely that overflow "uncouples" glycolysis from the TCA cycle. This point is substantiated by noting that the exchange fluxes are in the same range as the glucose uptake rate (Fig. 3A). I find this representation misleading in the sense that, when looking at the carbon balance, in the case of *E. coli* for example, the glucose uptake rate in Cmmol/gDW/h is three times higher than the acetate exchange fluxes in Cmmol/gDW/h. Consequently, contrary to what the picture in Fig. 3C suggests, not all acetyl CoA produced from glucose will leave the cell as acetate, a bit less than half will directly enter the TCA cycle. In other words, the uncoupling is only partial. I do understand the interest of the point the authors wish to make, but I think the claim and its graphical representation are too strong and need to be qualified.

2. The perspective of (partial) uncoupling means that overflow of acetate during growth on glucose on the one hand, and co-

utilization of glucose and acetate on the other, become phenomena that are quite similar. All the more so as the experiments reported in Fig. 3A were carried out in media containing a mixture of glucose and acetate. There is a substantial, older and recent literature on (the conditions of) the co-utilization of carbon sources (e.g., 10.15252/msb.20145537, 10.1038/s41467-019-09261-3, and 10.1098/rstb.1982.0055), which treats the latter as a phenomenon separate from overflow metabolism. Given that in the perspective proposed by the authors the two phenomena are fundamentally related, it is a bit surprising that the authors do not try to relate these two bodies of literature in more depth.

3. In recent years, there have been many studies that treat overflow metabolism from the perspective of proteome constraints, that is, these studies posit the existence of a trade-off between energy efficiency and protein costs under proteome constraints (e.g., 10.1038/nature15765, 10.1038/msb.2009.82, many FBA models of Warburg effect 10.1186/1752-0509-4-58, ...). This perspective is almost completely absent from the review and needs to be discussed in more detail.

4. The approach alluded to in the previous point attempts an explanation for the existence of overflow metabolism by viewing it as a consequence of proteomic constraints. The current manuscript is mostly descriptive and does not really adventure itself in the direction of giving explanatory hypotheses for the existence of overflow metabolism. The final sections propose that overflow metabolites may be "global regulators of metabolism" with "many positive roles for cells and populations" (p16). But not much evidence given for the role of overflow metabolites as "global regulators" and the positive roles detailed in the section "Overflow metabolites are not toxic per se..." mainly concern the possibility for overflow metabolites to be taken up again, which raises the question why overflow occurs in the first place. The question asked (Why overflow metabolism occurs?) is difficult, but one would expect at least a discussion in this review of (many) tentative answers given in the literature and maybe some new directions for research addressing this question from the perspective proposed by the authors.

5. Similar to the above point, I would be interested in learning about the view of the authors on the existence of the large variability in overflow phenotypes observed across individual strains of a species growing in the same conditions (e.g., for *E. coli*, see 10.1016/j.cels.2016.08.013 and the data collected in 10.7554/eLife.79815). Whereas in some strains a large proportion of carbon entering the cell is secreted, in other strains no net overflow occurs. Again, it may be difficult to give a conclusive answer to the question why this variability occurs, but a discussion of the observation and possible experimental leads would be interesting.

6. I find that the claims made by the authors are often formulated too strongly. I already made a remark on the notion of "uncoupling" above, but this applies to other points in the manuscript. Another example: "... clearly demonstrating that [acetate] is not toxic per se and ruling out previous hypotheses regarding its inhibitory effects on growth" (p12). The inhibitory effect of acetate on growth (at high concentrations) is not a "hypothesis", but an established fact and is not "ruled out" by the fact that it may also be a carbon source. It would be good if the authors could read through the manuscript to check the formulation of their claims and correct for the sometimes hyperbolic style.

Minor comments

1. It would be good to add Pdh to Fig. 1, given that this enzyme is mentioned in the text.
2. The control of the glycolytic flux on acetate overflow is indeed positive (p5), but only above a certain threshold (at least in the cases of *E. coli* and yeast). Below this threshold there is no overflow, so in this range the control coefficient is 0.

Reviewer #2:

Report on the manuscript "Overflow metabolism in bacterial, yeast, and mammalian cells: different names, same game" by Thomas Gosselin-Monplaisir and coworkers:

To say it up front, the title promises a comprehensive review illustrating the common principles of overflow metabolism in different organisms, but fails to keep it. Most importantly, the review ignores an important body of literature. On page 2, the authors state that "no overarching framework has yet been proposed" for overflow metabolism. This statement is simply not true. Several theoretical attempts have been made to explain the general phenomenon of overflow metabolism. Which one will turn out to be commonly accepted is a different question, but discussing these ideas was what I expected when starting to read the manuscript.

At least three independent ideas come to mind which definitely must be discussed in a review on overflow metabolism:

1. The authors themselves cite work by Terrence Hwa (see ref. 40), where capacity constraints are proposed as a major reason for the onset of overflow metabolism. While in this publication, the focus is on *E. coli* and corresponding experimental and theoretical evidence is provided, it is rather apparent that the proposed mechanism might serve as a general common principle. However, the main message of this paper is not even mentioned in the manuscript, which makes me wonder whether the authors actually carefully read and understood this important work.

2. Various constraint-based approaches come to the same conclusion. In particular noteworthy is the publication by de Groot et al., 2019, *Cellular and Molecular Life Sciences*, Vol. 77, No. 3: The common message of constraint-based optimization approaches: overflow metabolism is caused by two growth-limiting constraints. Whether or not capacity constraints are the actual cause of overflow of course remains to be seen, but not even mentioning these hypotheses in a review about overflow metabolism is negligent at the very least.

3. Another key publication in this area is Niebel et al., 2019, *Nature Metabolism*, Vol. 1, No. 1: An upper limit on Gibbs energy dissipation governs cellular metabolism. Here, an alternative hypothesis is proposed, namely that there is a principle upper limit of Gibbs free energy that can be dissipated by a cell, and that this constraints results in overflow metabolism. A rather involved thermodynamic genome-scale network model is used to support this hypothesis with modelling and experimental results.

4. In Saadat et al., 2020, *Entropy*, Vol. 22, No. 3: Thermodynamic Limits and Optimality of Microbial Growth, the authors present a minimal model that illustrates how this hypothesis of maximal Gibbs energy dissipation may explain overflow metabolism. While it is by no means clear whether this principle upper bound exists and what might cause it, it is an attractive hypothesis (as is the capacity constraint theory explored in the first two examples) with the capacity to provide a general theoretical framework to explain the phenomenon of overflow metabolism.

These are just some examples (which I happen to be most familiar with) of a considerable body of literature that discusses overflow metabolism from a general point of view with the attempts to provide theoretical explanations of general validity. Therefore the statement that no overarching framework has yet been proposed is clearly wrong. This omission is the greatest flaw of this manuscript and the reason why I recommend to reject its publication.

That said, however, I would also like to stress positive aspects of this review, first of all the fact that it is in general very well written. If the manuscript is rewritten to provide a fair and unbiased overview of existing theories, it could become a valuable contribution. To avoid the length to grow extensively, I recommend to remove the later sections, which do not really provide any valuable insight into the main topic (in particular "Controlling overflow metabolism for biotechnology" and "Overflow metabolism in health and disease" can easily be omitted, and the sections before can be considerably shortened, as they are also somewhat repetitive).

Other points:

Figure 1B: the color coding is ambiguous. Moreover, the experimental data presented raises some questions. Why, for example, do the cell numbers decrease in all three experiments?

p.4: "While the topology of overflow pathways has long been established for all microorganisms" - 'all' is certainly an exaggeration.

Section "The high reversibility of overflow metabolism uncouples carbon oxidation from nutrient uptake": this is the most interesting and strongest section of this review. In combination with other hypothesis on overflow metabolism, this could form the nucleus of a truly valuable review on overflow metabolism.

"Thermodynamic control": the authors repeatedly use the expression "thermodynamic control", including in the section heading on page 9. This expression requires a precise definition and explanation. If thermodynamic controls overflow fluxes, then this means - to my understanding - that alone thermodynamic gradients caused by concentration gradients determine these fluxes. In other words, only the concentrations of metabolites will determine whether and under which circumstances overflow metabolism is active. However, this is certainly not the case. If this were true knock-down, overexpression or IPTG-induced induction of enzymes should not affect overflow metabolism, which is in clear contradiction with the findings presented in Figure 2. Unless clearly defined what the authors mean by "thermodynamic control" I suggest to avoid this rather poorly defined term. It seems rather obvious that overflow is controlled by various factors, with thermodynamic gradients being only one of them, but kinetic regulation appears at least as important, if not more. Moreover, it is astonishing that in a review on thermodynamics and overflow metabolism key literature (see 3 and 4 above) is not even mentioned.

Response to Reviewers' comments

We thank the two reviewers for their interest in this work and their detailed feedback. We agree with all the comments and have carefully addressed them in the revised version of the manuscript. As suggested also, we have replaced the two last sections on potential applications of the identified principles with a new section that examines current theories on overflow metabolism, in light of the identified principles. We discuss how to drive research toward the identification of a unified theory of overflow metabolism. We feel this new section adds depth and scientific value, significantly strengthening the manuscript. We have also rephrased (unintended) overstatements.

We have tracked changes in the revised manuscript and reply point-by-point to the reviewers' comments below (reviewers' comments in black, our responses in blue).

Reviewer #1:

The manuscript by Gosselin-Monplaisir reviews the phenomenon of overflow metabolism, drawing upon results from bacteria, yeast, and mammals. The corresponding literatures, starting more than a century ago, have been often read in a disconnected way and the ambition of this review is to identify features that are generic across overflow metabolism in all of these different organisms. Maybe the most interesting novelty of the perspective that is proposed, supporting the claim of providing "fundamental insights" and "universal principles" (abstract), is the idea that overflow metabolites are not to be considered toxic waste, but nutrients and regulators of both cells generating the overflow and cells of other species in a microbial community.

The review is interesting, well-informed, and well-written, but can be improved and/or clarified on a number of points.

Major comments

1. Fig. 3C summarizes one of the main points of this review, namely that overflow "uncouples" glycolysis from the TCA cycle. This point is substantiated by noting that the exchange fluxes are in the same range as the glucose uptake rate (Fig. 3A). I find this representation misleading in the sense that, when looking at the carbon balance, in the case of *E. coli* for example, the glucose uptake rate in Cmmol/gDW/h is three times higher than the acetate exchange fluxes in Cmmol/gDW/h. Consequently, contrary to what the picture in Fig. 3C suggests, not all acetyl CoA produced from glucose will leave the cell as acetate, a bit less than half will directly enter the TCA cycle. In other words, the uncoupling is only partial. I do understand the interest of the point the authors wish to make, but I think the claim and its graphical representation are too strong and need to be qualified.

We thank reviewer 1 for raising this point. Figure 3C was designed to illustrate the conceptual shift introduced by the decoupling mechanism: rather than glucose simply being partitioned at the acetyl-CoA node between the TCA cycle and overflow pathways, (net) overflow and TCA fluxes can vary largely independently of glycolytic flux. We recognize that our initial explanation may not have been sufficiently clear and that the proposed

representation, which did not aim to be quantitative, could be misleading. To address this, we explicitly mention in the revised manuscript that the uncoupling is only partial, and we have modified panel 3C to clarify the graphical representation of this conceptual shift.

2. The perspective of (partial) uncoupling means that overflow of acetate during growth on glucose on the one hand, and co-utilization of glucose and acetate on the other, become phenomena that are quite similar. All the more so as the experiments reported in Fig. 3A were carried out in media containing a mixture of glucose and acetate. There is a substantial, older and recent literature on (the conditions of) the co-utilization of carbon sources (e.g., 10.15252/msb.20145537, 10.1038/s41467-019-09261-3, and 10.1098/rstb.1982.0055), which treats the latter as a phenomenon separate from overflow metabolism. Given that in the perspective proposed by the authors the two phenomena are fundamentally related, it is a bit surprising that the authors do not try to relate these two bodies of literature in more depth.

This comment reinforces our view that the outlined principles call for a fresh reanalysis of various aspects of cell physiology. We fully agree with reviewer 1 that the perspective of uncoupling provides a conceptual link between two apparently distinct phenomena: overflow metabolism and co-utilization of carbon sources. A comprehensive reexamination of the numerous theories about nutrient co-utilization (and determining whether these theories can be applied to overflow metabolites) is beyond the scope of this review, but we recognize the importance of highlighting this connection, which we now mention in the revised manuscript (page 18):

“This suggests that the production and the co-utilization of overflow metabolites are two faces of the same coin. However, studies on the co-utilization of different carbon sources (e.g., Hermsen et al, 2015; Wang et al, 2019; Harder & Dijkhuizen, 1982; Okano et al, 2019) typically treat overflow metabolism as a distinct phenomenon. Given their potential interconnection, future research should explore whether current theories of nutrient co-utilization could also apply to overflow metabolites.”

We thank reviewer 1 for raising this insightful point.

3. In recent years, there have been many studies that treat overflow metabolism from the perspective of proteome constraints, that is, these studies posit the existence of a trade-off between energy efficiency and protein costs under proteome constraints (e.g., 10.1038/nature15765, 10.1038/msb.2009.82, many FBA models of Warburg effect 10.1186/1752-0509-4-58, ...). This perspective is almost completely absent from the review and needs to be discussed in more detail.

Once again, this comment reinforces our view that the identified principles call for a fresh reanalysis of various aspects of cell physiology, including, of course, a critical reassessment of theories on the causes of overflow metabolism.

In the original version, we chose to focus on the potential applications of the empirical principles we identified, rather than on the causes of overflow metabolism. However, in light of the reviewers' comments, we recognize the value of presenting current theories on the causes of overflow, some of which were only briefly mentioned in the discussion of the

original manuscript. We develop our views on this topic in a new section, entitled "*Toward a unified theory of overflow metabolism*" (page 15), where we review prevailing theories on overflow metabolism. Although our personal concern was primarily with applications arising from the principles we outline, the fact that both reviewers emphasized the need to reinterpret the causes of overflow has convinced us to adopt their point of view. The absence of a comprehensive review of overflow theories in the literature also supports this change. No theory has yet been conclusively validated as a universal explanation for overflow metabolism, and how the proposed theories interrelate is unclear even within particular organisms. To address these gaps in the literature, the new section provides a perspective on how to drive research toward a unified theory of overflow metabolism.

4. The approach alluded to in the previous point attempts an explanation for the existence of overflow metabolism by viewing it as a consequence of proteomic constraints. The current manuscript is mostly descriptive and does not really adventure itself in the direction of giving explanatory hypotheses for the existence of overflow metabolism. The final sections propose that overflow metabolites may be "global regulators of metabolism" with "many positive roles for cells and populations" (p16). But not much evidence given for the role of overflow metabolites as "global regulators" and the positive roles detailed in the section "Overflow metabolites are not toxic per se..." mainly concern the possibility for overflow metabolites to be taken up again, which raises the question why overflow occurs in the first place. The question asked (Why overflow metabolism occurs?) is difficult, but one would expect at least a discussion in this review of (many) tentative answers given in the literature and maybe some new directions for research addressing this question from the perspective proposed by the authors.

As mentioned above, while the initial manuscript focused on the applicative value of the identified principles, we have replaced these two sections with a discussion on current theories of overflow metabolism and outlined new research directions to address this question. While this new section does not provide a definitive answer to the complex question of "why overflow metabolism occurs", it offers new insights into how this question can be effectively addressed.

5. Similar to the above point, I would be interested in learning about the view of the authors on the existence of the large variability in overflow phenotypes observed across individual strains of a species growing in the same conditions (e.g., for *E. coli*, see 10.1016/j.cels.2016.08.013 and the data collected in 10.7554/eLife.79815). Whereas in some strains a large proportion of carbon entering the cell is secreted, in other strains no net overflow occurs. Again, it may be difficult to give a conclusive answer to the question why this variability occurs, but a discussion of the observation and possible experimental leads would be interesting.

We agree with this comment, which once again confirms the broad impact of our findings. While a definitive answer would require dedicated studies and is beyond the scope of this review, we agree that this issue is relevant and of interest to readers. We now briefly discuss metabolic factors that could explain the differences observed between strains (page 6):

"This distributed control pattern, consistently observed across different organisms, may help explain the significant phenotypic diversity of overflow metabolism in different strains in that

it is linked to the regulation of glycolysis and the TCA cycle. For instance, flux measurements in seven yeast species have shown that a combination of low glycolytic fluxes with high TCA fluxes leads to higher growth rates and lower ethanol production (Christen & Sauer, 2011). Similarly, variability in overflow metabolism among natural and engineered E. coli strains can, at least in part, be attributed to differences in regulation of their glycolytic and TCA cycle activities (Fuentes et al, 2013; Castaño-Cerezo et al, 2015; Lozano Terol et al, 2019; Waegeman et al, 2011; Marisch et al, 2013; Monk et al, 2016; Baldazzi et al, 2023)."

We thank reviewer 1 for raising this important point.

6. I find that the claims made by the authors are often formulated too strongly. I already made a remark on the notion of "uncoupling" above, but this applies to other points in the manuscript. Another example: "... clearly demonstrating that [acetate] is not toxic per se and ruling out previous hypotheses regarding its inhibitory effects on growth" (p12). The inhibitory effect of acetate on growth (at high concentrations) is not a "hypothesis", but an established fact and is not "ruled out" by the fact that it may also be a carbon source. It would be good if the authors could read through the manuscript to check the formulation of their claims and correct for the sometimes hyperbolic style.

We apologize for the lack of clarity and any unintended overstatements. Our intention was not to suggest that the inhibitory effect of acetate on growth observed at high glycolytic flux is merely a hypothesis, as this is a well-established fact that we do not dispute.

In stating that "*[acetate] is not toxic per se and ruling out previous hypotheses regarding its inhibitory effects on growth*", we were specifically rejecting certain mechanistic explanations that have been proposed as a cause for the toxicity observed at high glycolytic flux. As discussed in our recent EMBO journal article (doi: 10.15252/embj.2022113079), we have found that acetate is not always toxic for *E. coli*, even at high concentration. In fact, at low glycolytic flux, the presence of 60 mM acetate enhances *E. coli* growth, and the presence of 100 mM acetate results in the same growth rate as observed in the absence of acetate. These recent findings contradict some of the mechanisms proposed to explain acetate's inhibitory effect on *E. coli* growth at high glycolytic flux.

We have carefully reviewed the manuscript to avoid any ambiguity in our statements.

Minor comments

1. It would be good to add Pdh to Fig. 1, given that this enzyme is mentioned in the text.

We have added Pdh to Fig. 1.

2. The control of the glycolytic flux on acetate overflow is indeed positive (p5), but only above a certain threshold (at least in the cases of *E. coli* and yeast). Below this threshold there is no overflow, so in this range the control coefficient is 0.

We agree with this comment and have clarified this point in the revised manuscript.

Reviewer #2:

Report on the manuscript "Overflow metabolism in bacterial, yeast, and mammalian cells: different names, same game" by Thomas Gosselin-Monplaisir and coworkers:

To say it up front, the title promises a comprehensive review illustrating the common principles of overflow metabolism in different organisms, but fails to keep it. Most importantly, the review ignores an important body of literature. On page 2, the authors state that "no overarching framework has yet been proposed" for overflow metabolism. This statement is simply not true. Several theoretical attempts have been made to explain the general phenomenon of overflow metabolism. Which one will turn out to be commonly accepted is a different question, but discussing these ideas was what I expected when starting to read the manuscript.

We apologize for the lack of clarity and understand the reviewer's disappointment at the lack of discussion on the underlying causes of overflow metabolism.

Our focus in the original manuscript was on highlighting the common operational principles of overflow metabolism—how it functions—and demonstrating that these principles are conserved across many organisms. To keep the review concise, we chose to emphasize the potential applications of the identified principles rather than investigating the causes of overflow metabolism. This is why we did not initially include a detailed discussion of existing theories of overflow.

In the original manuscript, we only briefly mentioned a few theories and noted that they should be re-evaluated in light of the identified principles—particularly in light of how microorganisms respond to overflow products, and we fully agree with the reviewers' suggestion to expand this discussion. In the revised manuscript, we now present the prevailing theories on overflow metabolism and provide a more detailed analysis of how these disconnected theories could be tested, refined, and integrated toward a unified theory of overflow (see the new section "*Toward a unified theory of overflow metabolism*", page 15).

We are also pleased that our findings raise fundamental questions for reviewer 2 regarding current theories, highlighting the need for deeper discussion, in line with the objective of this review, i.e. to prompt a re-examination of many aspects of overflow metabolism.

We have carefully reviewed and addressed all reviewer 2's comments. In particular, we have reformulated certain statements, expanded our discussion of existing theories, and explored avenues toward a potential unified theory of overflow metabolism.

We would like to thank reviewer 2 for their insightful comments, which have helped us improve the manuscript.

At least three independent ideas come to mind which definitely must be discussed in a review on overflow metabolism:

1. The authors themselves cite work by Terrence Hwa (see ref. 40), where capacity constraints are proposed as a major reason for the onset of overflow metabolism. While in this publication, the focus is on *E. coli* and corresponding experimental and theoretical evidence is provided, it is rather apparent that the proposed mechanism might serve as a general common principle. However, the main message of this paper is not even mentioned in the manuscript, which makes me wonder whether the authors actually carefully read and understood this important work.

2. Various constraint-based approaches come to the same conclusion. In particular noteworthy is the publication by de Groot et al., 2019, *Cellular and Molecular Life Sciences*, Vol. 77, No. 3: The common message of constraint-based optimization approaches: overflow metabolism is caused by two growth-limiting constraints. Whether or not capacity constraints are the actual cause of overflow of course remains to be seen, but not even mentioning these hypotheses in a review about overflow metabolism is negligent at the very least.

3. Another key publication in this area is Niebel et al., 2019, *Nature Metabolism*, Vol. 1, No. 1: An upper limit on Gibbs energy dissipation governs cellular metabolism. Here, an alternative hypothesis is proposed, namely that there is a principle upper limit of Gibbs free energy that can be dissipated by a cell, and that this constraints results in overflow metabolism. A rather involved thermodynamic genome-scale network model is used to support this hypothesis with modelling and experimental results.

4. In Saadat et al., 2020, *Entropy*, Vol. 22, No. 3: Thermodynamic Limits and Optimality of Microbial Growth, the authors present a minimal model that illustrates how this hypothesis of maximal Gibbs energy dissipation may explain overflow metabolism. While it is by no means clear whether this principle upper bound exists and what might cause it, it is an attractive hypothesis (as is the capacity constraint theory explored in the first two examples) with the capacity to provide a general theoretical framework to explain the phenomenon of overflow metabolism.

These are just some examples (which I happen to be most familiar with) of a considerable body of literature that discusses overflow metabolism from a general point of view with the attempts to provide theoretical explanations of general validity. Therefore the statement that no overarching framework has yet been proposed is clearly wrong. This omission is the greatest flaw of this manuscript and the reason why I recommend to reject its publication.

As mentioned above, our primary objective was to outline important empirical similarities between organisms, a topic that has not been systematically investigated in the literature, and not to analyze theories explaining the causes of overflow metabolism. We realize this should have been discussed in more detail.

We confirm that we have carefully read and—we think—understood Terrence Hwa's important work on *E. coli*, which we initially only briefly mentioned in the discussion of the original manuscript. We are also aware of the other compelling theories and articles referenced by reviewer 2.

As stated above, we now present our views on existing theories of overflow metabolism and explore how these might be combined into a single theory of overflow metabolism. We

discuss these theories in a new section, entitled *“Toward a unified theory of overflow metabolism.”*

For instance, a key consideration is that most of these theories are based on results from growth experiments performed in the absence of by-products, and it is therefore unclear how they account for the functional principles highlighted in this review—particularly reversibility, partial uncoupling, and co-utilization of overflow metabolites with glycolytic nutrients. We present and examine specific aspects of the proposed theories in the revised manuscript. Testing and integrating these disconnected theories will of course require dedicated studies and is beyond the scope of this review.

That said, however, I would also like to stress positive aspects of this review, first of all the fact that it is in general very well written. If the manuscript is rewritten to provide a fair and unbiased overview of existing theories, it could become a valuable contribution. To avoid the length to grow extensively, I recommend to remove the later sections, which do not really provide any valuable insight into the main topic (in particular "Controlling overflow metabolism for biotechnology" and "Overflow metabolism in health and disease" can easily be omitted, and the sections before can be considerably shortened, as they are also somewhat repetitive).

We have followed reviewer's recommendation by replacing the two sections *"Controlling overflow metabolism for biotechnology"* and *"Overflow metabolism in health and disease"* by a new section, *“Toward a unified theory of overflow metabolism”*.

Other points:

Figure 1B: the color coding is ambiguous. Moreover, the experimental data presented raises some questions. Why, for example, do the cell numbers decrease in all three experiments?

We have clarified the legend to avoid confusion (it is overflow metabolites that systematically decrease after glucose exhaustion, not cell numbers).

p.4: "While the topology of overflow pathways has long been established for all microorganisms" - 'all' is certainly an exaggeration.

We have changed “all” to “many”. We have also carefully checked the wording of our statements throughout the manuscript.

Section "The high reversibility of overflow metabolism uncouples carbon oxidation from nutrient uptake": this is the most interesting and strongest section of this review. In combination with other hypothesis on overflow metabolism, this could form the nucleus of a truly valuable review on overflow metabolism.

We thank reviewer 2 for their interest in this section. We hope they will also appreciate the new section presenting and discussing prevailing theories on overflow metabolism.

"Thermodynamic control": the authors repeatedly use the expression "thermodynamic

control", including in the section heading on page 9. This expression requires a precise definition and explanation.

We apologize for the lack of clarity, we now explain this term in more detail in the revised manuscript (page 10):

"The recent discovery that overflow is a highly reversible process has led to the suggestion that increasing the extracellular concentration of by-products could modulate the thermodynamic gradient of overflow pathways and, through this thermodynamic driving force, regulate their flux. This hypothesis of local thermodynamic control of overflow pathways has been tested in detail in E. coli (Enjalbert et al, 2017)."

If thermodynamic controls overflow fluxes, then this means - to my understanding - that alone thermodynamic gradients caused by concentration gradients determine these fluxes. In other words, only the concentrations of metabolites will determine whether and under which circumstances overflow metabolism is active.

The existence of thermodynamic control indeed means that "thermodynamic gradients caused by concentration gradients determine these fluxes," precisely as understood by reviewer 2. We have shown in *E. coli* that "the concentrations of metabolites determine whether and under which circumstances overflow metabolism is active", as also understood by reviewer 2, and the data presented in this review further support this view in yeast and mammalian cells.

However, we do not agree that the existence of thermodynamic control implies that fluxes are governed *solely* by thermodynamics. As stated in the section "Control of overflow fluxes is shared between overflow pathways, glycolysis, and the TCA cycle", we extensively describe several additional mechanisms that contribute to overflow flux regulation.

We clarify this point in the revised manuscript, stating that "*While local thermodynamic forces are key drivers of flux reversal and co-utilization of overflow metabolites, enzymatic regulation is equally critical, indicating that overflow metabolism operates in a multi-layered regulatory system*" (page 16). We also look further into the cause of overflow metabolism, including theories on thermodynamic control of overflow fluxes in the new section "*Toward a unified theory of overflow metabolism*", and in the discussion section, we explicitly acknowledge that "*Moreover, overflow metabolism is likely governed by a combination of thermodynamic, enzymatic, biophysical, and evolutionary constraints. Future research should focus on integrating existing theories and testing them in light of the outlined principles. The regulatory programs triggered by overflow metabolites will also need to be identified*" (page 19).

However, this is certainly not the case. If this were true knock-down, overexpression or IPTG-induced induction of enzymes should not affect overflow metabolism, which is in clear contradiction with the findings presented in Figure 2. Unless clearly defined what the authors mean by "thermodynamic control" I suggest to avoid this rather poorly defined term. It seems rather obvious that overflow is controlled by various factors, with thermodynamic gradients being only one of them, but kinetic regulation appears at least as important, if not more.

We apologize for the lack of clarity in the meaning of “thermodynamic control”. We now state that *“The recent discovery that overflow is a highly reversible process has led to the suggestion that increasing the extracellular concentration of by-products could modulate the thermodynamic gradient of overflow pathways and, through this thermodynamic driving force, regulate their flux. This hypothesis of local thermodynamic control of overflow pathways has been tested in detail in E. coli”* (page 10).

Moreover, the fact that thermodynamic control has been validated in *E. coli* (in studies published in 2017, 2021 and 2023) does not mean that fluxes are determined *solely* by this mechanism. We explicitly state that overflow fluxes are regulated by multiple factors, including enzyme activities, as discussed at length in the section entitled *“Control of overflow fluxes is shared between overflow pathways, glycolysis, and the TCA cycle”* and supported by the findings presented in Figure 2. We now state this explicitly in the added section , *“Toward a unified theory of overflow metabolism”*. We also state in the discussion that *“overflow metabolism is likely governed by a combination of thermodynamic, enzymatic, biophysical, and evolutionary constraints”* (page 19).

Moreover, it is astonishing that in a review on thermodynamics and overflow metabolism key literature (see 3 and 4 above) is not even mentioned.

In the revised manuscript, we present the key literature on thermodynamic constraint at the cell level (including the publications discussed in points 3 and 4). We also clarify the difference between local thermodynamic control of overflow pathways and global thermodynamic constraint at the cell level (notably in the new section, *“Toward a unified theory of overflow metabolism”*).

3rd Jul 2025

Manuscript Number: MSB-2024-12779R

Title: Overflow metabolism in bacterial, yeast, and mammalian cells: different names, same game

Dear Dr Millard,

Thank you for the submission of your revised Review to Molecular Systems Biology.

Please find below the two sets of comments I have now received regarding the re-review of your piece. As you will see, both referees are positive about its timeliness and suitability for publication. However, Reviewer 1 has some remaining suggestions that should be addressed prior to publication.

When submitting your revised manuscript, please also address the following formatting requests:

- 1) Please remove all figures from main manuscript file and leave only main figure legends placed after the References. We will send Figure 6 to our graphic artist who will edit/re-draw it for style and clarity. The graphic artist will contact you directly for your approval on the final figure.
- 2) Please include up to 5 keywords
- 3) Author contributions: Please remove it from the manuscript and specify author contributions in our submission system. CRediT has replaced the traditional author contributions section because it offers a systematic machine-readable author contributions format that allows for more effective research assessment. You are encouraged to use the free text boxes beneath each contributing author's name to add specific details on the author's contribution. More information is available in our guide to authors:
<https://www.embopress.org/page/journal/17574684/authorguide#authorshipguidelines>
- 4) Funding: Please note that funding information should be given in the "Acknowledgements" section (not in its own separate section).
- 5) There are callouts for Figures 1C-D, but there are only Panels A-B in the figure.
- 6) Please ensure that the section order of the manuscript is corrected to the following: Title page - Abstract & Keywords - Introduction - Results/Review subsections - Conclusions - Disclosure and Competing Interests Statement - Acknowledgements - References - Figure Legends
- 7) As part of the EMBO Publications transparent editorial process initiative (see our policy here: https://www.embopress.org/transparent-process#Review_Process), Molecular Systems Biology will publish online a Peer Review File (PRF) to accompany accepted manuscripts. This file will be published in conjunction with your paper and will include the anonymous referee reports, your point-by-point response and all pertinent correspondence relating to the manuscript. Let us know whether you agree with the publication of the PRF and as here, if you want to remove or not any figures from it prior to publication.
- 8) Please provide a point-by-point letter INCLUDING my comments as well as the reviewer's reports and your detailed responses (as Word file).

If you have any questions, please don't hesitate to ask. I look forward to seeing the revised manuscript.

Yours sincerely,

Poonam Bheda, PhD
Senior Scientific Editor
Molecular Systems Biology

Reviewer #1:

The authors have made a real effort to improve the paper in response to my comments on the first version of the manuscript. The main changes are the removal of the sections on the potential role of overflow metabolism in biotechnology, health, and diseases (a good decision) and their replacement by a section discussing the empirical "principles" identified by the authors in the context of existing theoretical frameworks for the explanation of overflow metabolism (something that was clearly missing).

The discussion of previous theoretical frameworks in the section "Toward a unified theory of overflow metabolism" could be further clarified.

A first point is the specific status of the "alternative theory" developed by the authors (p16-17). On the one hand, this seems to

concern the assumption that "overflow metabolism is largely governed by thermodynamic gradients between cells and their environment". But later in the same paragraph the authors modulate this assumption by positing that enzyme regulation must also play a role, as inferred from a model including regulation of glycolysis and the TCA cycle by acetate in *E. coli*. They conclude by saying that "key predictions of this theory have been validated in *E. coli*...". I guess that "this theory" refers to the model the authors developed? But such a model accounting for a specific situation in *E. coli* can hardly be compared to the theories with a more general scope the authors discussed before. More generally, saying that overflow metabolism involves thermodynamic gradients and regulation is not really an explanation of "why overflow occurs" [p15].

It would be more fruitful, I believe, if the authors used their "principles", which are basically empirical generalization from observations of overflow metabolism and related phenomena like co-utilization of substrates in different organisms, as criteria for evaluating existing (and future) theoretical frameworks, as a set of observations that any theory should be able to explain. The authors are right that none of theoretical frameworks they discuss are capable of accounting for the entire range of phenomena they relate to overflow in this review. It could even be considered to list these principles (empirical generalizations) at this stage of the paper, as a summary of the main contribution of their work (a table?).

Minor comments:

p11: "A kinetic model predicted that decreasing the glycolytic flux would reduce the acetate flux in the absence of acetate, but that the same perturbation would enhance acetate utilization in the presence of acetate". I find this phrase unclear, notably the use of "but". The two statements are not opposed, they are complementary because the experiments were carried out under completely different experimental conditions (without/with supplementation of acetate in the medium). Maybe replace "but" by "and even"?

p11: The two phrases immediately afterwards are contradictory. In the first phrase, acetate flux is said to control glycolytic flux, in the second phrase the opposite. It seems more logical to say that the glycolytic flux controls acetate overflow, especially since the experiments carried out to confirm this point manipulated the glycolytic flux.

p13: "Overflow metabolites are often referred to as "stress factors", but recognizing these compounds as global regulators is crucial...". Again, I don't understand the use of "but" here. The fact that they are global regulators is simply a more precise description of their role as stress factors?

p14: "Unexpectedly, acetate even boosts *E. coli* growth when the glycolytic flux is reduced, clearly demonstrating that it is not toxic per se and ruling out previous mechanistic explanations for its inhibitory effect on growth". I don't get the point why this is "unexpected". *E. coli* is perfectly capable of growing on acetate, as the authors know, in the absence of other (glycolytic) carbon sources. And I don't see why this rules out previous explanations for the growth inhibitory effect of acetate, which remain valid in situations other than those considered here (reduced influx of a glycolytic substrate). I think that the authors want to say that acetate does not always, when supplemented to a preferred glycolytic substrate, have a growth inhibitory effect. The authors should more precisely formulate their point.

p15: "Zhuang et al. propose that acetate overflow in *E. coli* results from competition for membrane space between glucose transporters and respiratory chain components, forcing cells to favor overflow when membrane space is limiting". This phrase is not clear when it is not explained that ATP production by fermentation (accompanied by overflow) requires less membrane space than respiration.

p15: "Basan et al. argue that bacterial oxidative metabolism, while energy-efficient, demands substantial enzyme investment because of its slow reaction rates, ...". I did not check the Basan paper for this claim of slow reaction rates, but my understanding was that respiration requires large protein complexes and long pathways with many enzymes for ATP generation as compared to fermentation, that is, a larger protein investment per unit of flux. This is not the same as "slow reaction rates" inherent to respiratory enzymes.

p16: "Other studies have proposed that overflow confers a competitive advantage by enabling cells to rapidly consume shared resources before competing species do". This is not an explanation of overflow metabolism, as it makes the prior assumption that overflow must occur for rapid consumption of the shared resource, which is precisely the phenomenon to be explained...

p17: "Similarly, the positive effect of overflow metabolites on growth undermines theories...". The positive effect "in certain conditions"!

p18: "their role as nutrient or by-product determined by their concentration". I think the authors refer here to the thermodynamic control of overflow pathways, and therefore the concentration of overflow metabolites "relative" to internal metabolites?

p18: "overflow metabolites should not be considered stress factors". Should not "always" be considered (they are stress factors in certain conditions...).

Reviewer #2:

I would like to commend the authors to their revised manuscript. In my opinion they have answered all questions and concerns in a clear and convincing manner and adapted the manuscript according to the suggestions. The manuscript has greatly improved and should be published in this revised form.

Editorial comments:

1) Please remove all figures from main manuscript file and leave only main figure legends placed after the References. We will send Figure 6 to our graphic artist who will edit/re-draw it for style and clarity. The graphic artist will contact you directly for your approval on the final figure.

We have implemented the requested changes. We have received and approved the new Figure 6; however, we are not providing the file for this figure in the online submission system, as we do not have the final version. Please contact the team you commissioned for it.

2) Please include up to 5 keywords

We have added keywords.

3) Author contributions: Please remove it from the manuscript and specify author contributions in our submission system. CRediT has replaced the traditional author contributions section because it offers a systematic machine-readable author contributions format that allows for more effective research assessment. You are encouraged to use the free text boxes beneath each contributing author's name to add specific details on the author's contribution. More information is available in our [guide](https://www.embopress.org/page/journal/17574684/authorguide#authorshipguidelines) to authors: <https://www.embopress.org/page/journal/17574684/authorguide#authorshipguidelines>

We have implemented the requested changes.

4) Funding: Please note that funding information should be given in the "Acknowledgements" section (not in its own separate section).

We have implemented the requested changes.

5) There are callouts for Figures 1C-D, but there are only Panels A-B in the figure.

This has been corrected, we have removed callouts for Figures 1C-D.

6) Please ensure that the section order of the manuscript is corrected to the following: Title page - Abstract & Keywords - Introduction - Results/Review subsections - Conclusions - Disclosure and Competing Interests Statement - Acknowledgements - References - Figure Legends

We confirm the section order of the manuscript is correct.

7) As part of the EMBO Publications transparent editorial process initiative (see our policy here: https://www.embopress.org/transparent-process#Review_Process), Molecular Systems Biology will publish online a Peer Review File (PRF) to accompany accepted manuscripts. This file will be published in conjunction with your paper and will include the anonymous referee reports, your point-by-point response and all pertinent correspondence relating to the manuscript. Let us know whether you agree with the publication of the PRF and as here, if you want to remove or not any figures from it prior to publication.

We agree with publication of the PRF, which is a highly valuable material accompanying scientific publications. We strongly support this initiative.

8) Please provide a point-by-point letter INCLUDING my comments as well as the reviewer's reports and your detailed responses (as Word file).

We have provided this file.

Reviewer #1:

The authors have made a real effort to improve the paper in response to my comments on the first version of the manuscript. The main changes are the removal of the sections on the potential role of overflow metabolism in biotechnology, health, and diseases (a good decision) and their replacement by a section discussing the empirical "principles" identified by the authors in the context of existing theoretical frameworks for the explanation of overflow metabolism (something that was clearly missing).

We thank Reviewer 1 for their positive comments on the revised manuscript.

The discussion of previous theoretical frameworks in the section "Toward a unified theory of overflow metabolism" could be further clarified.

A first point is the specific status of the "alternative theory" developed by the authors (p16-17). On the one hand, this seems to concern the assumption that "overflow metabolism is largely governed by thermodynamic gradients between cells and their environment". But later in the same paragraph the authors modulate this assumption by positing that enzyme regulation must also play a role, as inferred from a model including regulation of glycolysis and the TCA cycle by acetate in *E. coli*.

The two assumption of a thermodynamic control and of a metabolic control of overflow fluxes are not mutually exclusive. Both mechanisms control overflow fluxes, as shown based on experimental data in the sections "*Control of overflow fluxes is shared between overflow pathways, glycolysis, and the TCA cycle*", "*Thermodynamic control of overflow fluxes can lead to co-consumption of overflow metabolites with glycolytic nutrients*", and "*Low uptake of glycolytic nutrients enhances co-utilization of overflow metabolites*". Saying that enzyme regulation plays a role in overflow flux control does not "modulate" the fact that the flux is also controlled thermodynamically.

They conclude by saying that "key predictions of this theory have been validated in *E. coli*...". I guess that "this theory" refers to the model the authors developed? But such a model accounting for a specific situation in *E. coli* can hardly be compared to the theories with a more general scope the authors discussed before.

We are not sure what the reviewer means by "more general scope". If this refers to the range of situations that can be explained by the theories, we do not consider that some theories have a broader scope than others, although their scopes have been validated under different conditions.

The theory developed in *E. coli* was initially established in specific situations (high glycolytic flux) and was supported by a metabolic model. The model predictions – and thus the underlying theory – have been validated in a variety of contexts: across a broad range of glycolytic fluxes, on different glycolytic substrates, in both wild-type and mutant strains, in response to chemical perturbations, and over a wide range of acetate concentrations. We therefore consider that this theory remains valid in a general scope, not only in "specific situations."

In contrast, the other theories have been proposed and tested only in the absence of acetate, and not in the presence of acetate – thus within more specific conditions under which key phenotypes are not observed. Nevertheless, we do not consider these theories to have a narrower scope, since it is currently unknown whether they could explain *E. coli* responses in the presence of acetate.

Overall, the different theories have simply been tested under different conditions, and none can be said to have a definitively broader scope than the others. As discussed in the manuscript and

recognized by Reviewer 1, assessing the validity of these theories across a wider range of conditions remains to be done.

More generally, saying that overflow metabolism involves thermodynamic gradients and regulation is not really an explanation of "why overflow occurs" [p15].

Thermodynamic gradients explain the mechanistic basis for why overflow occurs in the absence of overflow metabolites (i.e., the thermodynamic gradient drives the flux toward the production of overflow metabolites), and for why overflow is reduced or even reversed in their presence (i.e., the thermodynamic gradient drives the flux toward the utilization of overflow metabolites). This is demonstrated based on experimental data and discussed in details in the section "*Thermodynamic control of overflow fluxes can lead to co-consumption of overflow metabolites with glycolytic nutrients*".

Perhaps what the reviewer is questioning is the biological role or physiological relevance of this mechanism. It has been shown that such thermodynamic control enables *E. coli* to rapidly adapt to changes in the availability of acetate (Enjalbert et al., Scientific report, 2017; Millard et al., eLife, 2021) and glycolytic substrates (Millard et al., EMBO J, 2023).

To address this point explicitly, we have amended the manuscript to state:

"Moreover, while thermodynamic gradients and enzymatic regulation explains why overflow occurs and offer insight into the advantage of an active overflow pathway under fluctuating nutritional environments (Enjalbert et al, 2017; Millard et al, 2023), the physiological relevance of the observed phenotypes should be further explored."

This clarification ensures that both the mechanistic foundation and the adaptive significance of the proposed theory are clearly communicated.

It would be more fruitful, I believe, if the authors used their "principles", which are basically empirical generalization from observations of overflow metabolism and related phenomena like co-utilization of substrates in different organisms, as criteria for evaluating existing (and future) theoretical frameworks, as a set of observations that any theory should be able to explain. The authors are right that none of theoretical frameworks they discuss are capable of accounting for the entire range of phenomena they relate to overflow in this review. It could even be considered to list these principles (empirical generalizations) at this stage of the paper, as a summary of the main contribution of their work (a table?).

We agree with Reviewer 1 that the highlighted principles could serve as "criteria for evaluating existing (and future) theoretical frameworks." In line with this comment, we state:

"The principles highlighted in this study can serve as criteria for evaluating existing and future theories, providing a set of observations that any robust explanation should account for."

Minor comments:

p11: "A kinetic model predicted that decreasing the glycolytic flux would reduce the acetate flux in the absence of acetate, but that the same perturbation would enhance acetate utilization in the presence of acetate". I find this phrase unclear, notably the use of "but". The two statements are not opposed, they are complementary because the experiments were carried out under completely

different experimental conditions (without/with supplementation of acetate in the medium). Maybe replace "but" by "and even"?

We have replaced "but" by "and".

p11: The two phrases immediate afterwards are contradictory. In the first phrase, acetate flux is said to control glycolytic flux, in the second phrase the opposite. It seems more logical to say that the glycolytic flux control acetate overflow, especially since the experiments carried out to confirm this point manipulated the glycolytic flux.

We apologize for this formulation error and thank reviewer 1 for raising this point. We have corrected the first sentence: "This negative control of the acetate flux by glycolysis in the presence of acetate was experimentally confirmed by lowering the glycolytic flux using different approaches."

p13: "Overflow metabolites are often referred to as "stress factors", but recognizing these compounds as global regulators is crucial...". Again, I don't understand the use of "but" here. The fact that they are global regulators is simply a more precise description of their role as stress factors?

We have split this sentence and removed the word "but." We would like to highlight the distinction between these terms: "stress factors" implies a negative effect of the compounds on cell physiology, whereas "regulators" emphasizes a cellular response that is not always detrimental (as demonstrated by the positive effects of overflow metabolites on cell physiology, detailed in the section "Overflow metabolites are not toxic per se and can be beneficial to cells").

p14: "Unexpectedly, acetate even boosts *E. coli* growth when the glycolytic flux is reduced, clearly demonstrating that it is not toxic per se and ruling out previous mechanistic explanations for its inhibitory effect on growth". I don't get the point why this is "unexpected". *E. coli* is perfectly capable of growing on acetate, as the authors know, in the absence of other (glycolytic) carbon sources. And I don't see why this rules out previous explanations for the growth inhibitory effect of acetate, which remain valid in situations other than those considered here (reduced influx of a glycolytic substrate). I think that the authors want to say that acetate does not always, when supplemented to a preferred glycolytic substrate, have a growth inhibitory effect. The authors should more precisely formulate their point.

We have removed the word "Unexpectedly" and softened this sentence:

"Acetate even boosts *E. coli* growth when the glycolytic flux is reduced, clearly demonstrating that it is not always toxic and questioning previous mechanistic explanations for its inhibitory effect on growth (Millard et al., 2023)."

The rationale for why the beneficial response of *E. coli* to acetate rules out previous explanations for its growth-inhibitory effect is detailed in the discussion of the referenced publication (Millard et al., EMBO J, 2023):

"Importantly, these results also weaken several hypotheses that have been proposed to explain the apparent "toxicity" of acetate for microbial growth. Indeed, perturbations of the proton gradient, of the anion composition of the cell, or of methionine biosynthesis, would have a monotonic impact on growth. The slight but global activating effect of acetate on the transcriptome at low glycolytic flux, which correlates with the growth response, also contrasts with its inhibitory effect at high glycolytic flux. The results of this study indicate that mechanisms that only explain growth inhibition at high glycolytic flux are inadequate. Likewise, the recent suggestion that gluconeogenic substrates such as acetate enhance *E. coli* growth on glycolytic substrates (as observed for mixtures of oxaloacetate

with glycerol, xylose or fucose; Okano et al, 2020) also fails to capture the non-monotonic growth response of *E. coli* to acetate.”

p15: "Zhuang et al. propose that acetate overflow in *E. coli* results from competition for membrane space between glucose transporters and respiratory chain components, forcing cells to favor overflow when membrane space is limiting". This phrase is not clear when it is not explained that ATP production by fermentation (accompanied by overflow) requires less membrane space than respiration.

We agree with reviewer’s explanation and have reformulated this sentence:

“Zhuang et al. propose that acetate overflow in *E. coli* results from competition for membrane space between glucose transporters and respiratory chain components. ATP production by the cytosolic overflow pathway requires less membrane space than respiration, forcing cells to favor overflow when membrane space is limiting (Zhuang et al, 2011).”

p15: "Basan et al. argue that bacterial oxidative metabolism, while energy-efficient, demands substantial enzyme investment because of its slow reaction rates, ...". I did not check the Basan paper for this claim of slow reaction rates, but my understanding was that respiration requires large protein complexes and long pathways with many enzymes for ATP generation as compared to fermentation, that is, a larger protein investment per unit of flux. This is not the same as "slow reaction rates" inherent to respiratory enzymes.

We agree with reviewer’s comment and have reformulated this sentence:

“Overflow metabolism may also be driven by metabolic economics. Basan et al. argue that bacterial oxidative metabolism requires substantial enzyme investment to produce ATP, whereas overflow metabolism, though less efficient for ATP production, requires fewer enzymes (Basan et al, 2015). This trade-off between energy production efficiency and enzyme cost could explain the emergence of overflow in *E. coli* (Basan et al, 2015).”

p16: "Other studies have proposed that overflow confers a competitive advantage by enabling cells to rapidly consume shared resources before competing species do". This is not an explanation of overflow metabolism, as it makes the prior assumption that overflow must occur for rapid consumption of the shared resource, which is precisely the phenomenon to be explained...

We agree that this is not a mechanistic explanation of overflow metabolism. However, as stated at the beginning of this paragraph, all of the aforementioned theories assume that overflow metabolism emerged as a growth-optimization strategy, an evolutionary assumption that is both strong and controversial. In this paragraph, we discuss alternative evolutionary explanations that could confer an advantage to overflow metabolism, thereby providing some clues to its biological relevance.

p17: "Similarly, the positive effect of overflow metabolites on growth undermines theories...". The positive effect *in certain conditions*!

We have added “in certain conditions”.

p18: "their role as nutrient or by-product determined by their concentration". I think the authors refer here to the thermodynamic control of overflow pathways, and therefore the concentration of overflow metabolites *relative* to internal metabolites?

Though experimental data clearly demonstrate that the role of overflow metabolites as nutrient or by-product is determined by their extracellular concentration (see section “Thermodynamic control of overflow fluxes can lead to co-consumption of overflow metabolites with glycolytic nutrients” and the data shown in Figure 4), we agree with reviewer’s comment. We now mention in the dedicated section “Thermodynamic control of overflow fluxes can lead to co-consumption of overflow metabolites with glycolytic nutrients”:

“These data support our hypothesis that overflow fluxes in all organisms are largely controlled at the thermodynamic level by the concentration of the respective by-products (relative to the concentration of internal metabolites).”

p18: "overflow metabolites should not be considered stress factors". Should not "always" be considered (they are stress factors in certain conditions...).

We have added “always”.

Reviewer #2:

I would like to commend the authors to their revised manuscript. In my opinion they have answered all questions and concerns in a clear and convincing manner and adapted the manuscript according to the suggestions. The manuscript has greatly improved and should be published in this revised form.

We thank Reviewer 2 for their positive feedback.

22nd Aug 2025

Dear Dr Millard,

Thank you for submitting the revised version of your review, I am pleased to let you that it has been accepted for publication in Molecular Systems Biology.

Your manuscript will be processed for publication by EMBO Press. It will be copy edited and you will receive page proofs prior to publication. You will soon be contacted by Springer Nature to sign your publishing license. When you login to the customer service website, please use the following token to waive the article publication charges: MTUWMDY2NZAYOA

Should you experience any difficulty, please email publishing@embo.org.

If you are planning a Press Release on your article, please get in contact with embo_production@springernature.com as early as possible in order to coordinate publication and release dates.

Sincerely,

Poonam Bheda, PhD
Scientific Editor
Molecular Systems Biology